# BLOS1 mediates kinesin switch during endosomal recycling of LDL receptor

Chang Zhang[1,2,3], Chanjuan Hao[2], Guanghou Shui[1]*, Wei Li[2]*

[1]State Key Laboratory of Molecular Developmental Biology, Institute of Genetics and Developmental Biology, Chinese Academy of Sciences, Beijing, China; [2]Beijing Key Laboratory for Genetics of Birth Defects, Beijing Pediatric Research Institute; MOE Key Laboratory of Major Diseases in Children; Genetics and Birth Defects Control Center, Beijing Children's Hospital, Capital Medical University, National Center for Children's Health, Beijing, China; [3]University of Chinese Academy of Sciences, Beijing, China

**Abstract** Low-density lipoprotein receptor (LDLR) in hepatocytes plays a key role in plasma clearance of circulating LDL and in whole body cholesterol homeostasis. The trafficking of LDLR is highly regulated in clathrin-dependent endocytosis, endosomal recycling and lysosomal degradation. Current studies focus on its endocytosis and degradation. However, the detailed molecular and cellular mechanisms underlying its endosomal recycling are largely unknown. We found that BLOS1, a shared subunit of BLOC-1 and BORC, is involved in LDLR endosomal recycling. Loss of BLOS1 leads to less membrane LDLR and impairs LDL clearance from plasma in hepatocyte-specific BLOS1 knockout mice. BLOS1 interacts with kinesin-3 motor KIF13A, and BLOS1 acts as a new adaptor for kinesin-2 motor KIF3 to coordinate kinesin-3 and kinesin-2 during the long-range transport of recycling endosomes (REs) to plasma membrane along microtubule tracks to overcome hurdles at microtubule intersections. This provides new insights into RE's anterograde transport and the pathogenesis of dyslipidemia.

*For correspondence:
ghshui@genetics.ac.cn (GS);
liwei@bch.com.cn (WL)

**Competing interests:** The authors declare that no competing interests exist.

## Introduction

Low-density lipoprotein receptor (LDLR) is essential for cellular uptake of cholesterol-carrying low-density lipoproteins (LDL) and plays a crucial role in cholesterol homeostasis in mammals (*Goldstein and Brown, 2009*; *Goldstein and Brown, 2015*). After the binding of LDL on the cell surface, LDLR enters early endosomes through clathrin-mediated endocytosis (CME) and is further recycled to the cell membrane by the RAB4-dependent fast recycling pathway or the RAB11A-dependent slow recycling pathway (*Ullrich et al., 1996*; *van der Sluijs et al., 1992*; *Wijers et al., 2015*). In the slow recycling pathway, receptors are first sorted into the perinuclear endocytic recycling compartment (ERC) and then transported to the cell periphery by recycling endosomes (REs) (*Naslavsky and Caplan, 2018*; *Yamashiro et al., 1984*). Due to the long distance in trafficking, the slow recycling pathway is subjected to complex regulation. Factors including RAB11A effectors (RAB11-FIPs) and Eps15-homology domain-containing proteins (EHDs) participate in the trafficking from early endosome (EE) to the ERC (*Horgan and McCaffrey, 2009*; *Naslavsky and Caplan, 2011*).

N-kinesins, the motor proteins that drive anterograde transport on microtubules, have also been reported to function in slow recycling (*Hirokawa et al., 2009*). A kinesin-3 member KIF13A cooperates with RAB11A to generate and drive the peripheral transport of RE tubules (*Delevoye et al., 2014*; *Nakagawa et al., 2000*). In addition, KIF13A is essential for cargo delivery from RE to maturing melanosomes in melanocytes (*Delevoye et al., 2009*). KIF16B, another member of kinesin-3, functions in regulating the motility of early endosomes and degradation of EGF receptor, and in the

transcytosis of transferrin receptor (TfR) in polarized epithelial cells (*Hoepfner et al., 2005*; *Perez Bay et al., 2013*). A kinesin-2 member KIF3B has also been implicated in the recycling of TfR through the interaction with RAB11-FIP5 (*Schonteich et al., 2008*). Whether and how these different kinesins coordinate in the transport of REs are largely unknown.

BLOS1 is a shared subunit of BLOC-1 (biogenesis of lysosome-related organelles complex-1) and BORC (BLOC-one-related complex) (*Scott et al., 2018*). BLOC-1 is required for the biogenesis of cell-type-specific lysosome-related organelles, such as melanosomes in melanocytes and dense granules in platelets (*Bowman et al., 2019*; *Wei and Li, 2013*). BORC recruits ARL8 and couples lysosomes to kinesins for anterograde transport on specific microtubule tracks, and mediates the trafficking of synaptic vesicle precursors in neurons (*Guardia et al., 2016*; *Niwa et al., 2017*). There is increasing evidence that BLOC-1 is involved in the formation of endosomal tubular structures through cooperation with microtubule- and actin-associated machineries (*Delevoye et al., 2016*; *Ryder et al., 2013*).

Here, we show that BLOS1 regulates LDLR recycling in hepatocytes. BLOS1 acts as an adaptor of kinesin-2 and coordinates kinesin-2 and kinesin-3 in the anterograde transport of REs. Dysfunction of kinesin-2 or BLOS1 results in impaired trafficking of REs and KIF13A-positive tubular structures characterized by impassability at specific microtubule-microtubule intersections. Consequently, liver-specific *Bloc1s1* knockout mice exhibit reduced LDLR in the liver and elevated plasma LDL levels due to alternative lysosome degradation of LDLR after impaired endocytic recycling.

## Results

### Liver-specific knockout of *Bloc1s1* in mice leads to abnormal lipid metabolism

Previous studies have shown that constitutive knockout of *Bloc1s1* in mice produces embryonic lethality (*Scott et al., 2014*; *Zhang et al., 2014*). To study the function of BLOS1 in liver lipid metabolism, we generated a conditional knockout mouse mutant (cKO mice) by crossing *Bloc1s1* floxed mice (loxp mice) with Albumin (*Alb*)-cre mice, while littermates lacking *cre* gene served as control group (*Figure 1—figure supplement 1a*). Deletion of *Bloc1s1* was confirmed in hepatocytes isolated from cKO mouse liver by genomic PCR (*Figure 1—figure supplement 1b*). Because antibody to BLOS1 was not applicable, we used two other BLOC-1 subunits (Pallidin and Dysbindin) to monitor the loss of BLOS1 as the depletion of BLOS1 leads to the destabilization of other BLOC-1 subunits (*Zhang et al., 2014*). Indeed, the content of both subunits was significantly reduced in cKO mouse livers and hepatocytes, suggesting the depletion of BLOS1 (*Figure 1a,b*).

To investigate whether BLOS1 deficiency could affect liver lipid droplet content, we performed Oil Red O staining on frozen sections of mouse livers and found that cKO mice had fewer lipid droplets in the liver when fed on chow diet, and the accumulation of lipid droplets after starvation was largely inhibited in cKO mouse liver (*Figure 1c–e*). Considering that many important constituents of plasma are secreted by liver, we examined plasma samples of cKO mice and their control littermates by non-reduced SDS-PAGE analysis. Although there were no changes in several abundant plasma proteins, such as albumin and transferrin, a protein band at about 34 kD was consistently increased in plasma samples from cKO mice (*Figure 1f*). Mass spectrometry identified this protein as apolipoprotein E (ApoE) (*Figure 1—figure supplement 1c*), and immunoblotting confirmed the increase of ApoE in cKO mice plasma (*Figure 1g*).

ApoE is a core protein component of very-low-density lipoprotein (VLDL) and high-density lipoprotein (HDL). For this reason, lipoproteins in plasma samples from cKO and loxp mice were prestained with Sudan Black B and then separated by gradient native polyacrylamide gel electrophoresis. Using lipoproteins purified from pooled mouse plasma by sequential ultracentrifugation, we determined different lipoprotein bands in native gels (*Figure 1—figure supplement 1d*, left). We observed that in cKO mouse plasma, VLDL level was decreased, LDL was increased, and HDL had a tendency to increase (*Figure 1h*). Native gels subjected to Oil Red O staining also showed similar results (*Figure 1—figure supplement 1d*, right). The reduction of VLDL in cKO plasma apparently could not account for the increase of ApoE content. We wondered whether ApoE is increased in the HDL fraction of lipoproteins. To address this point, we separated unstained lipoproteins from cKO and loxp mice using native gels, and then performed a second dimensional SDS-PAGE and

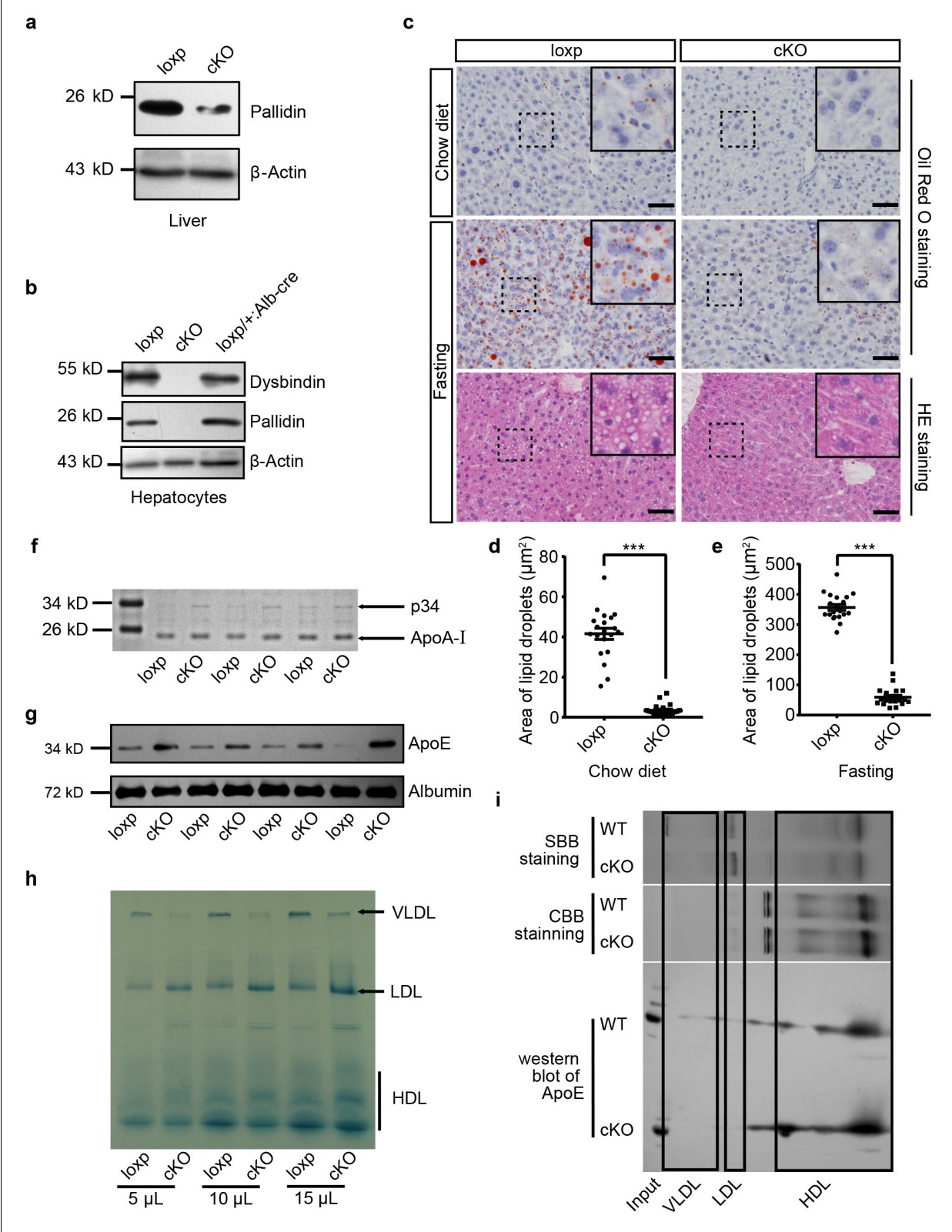

**Figure 1.** Mice with hepatocyte-specific deletion of BLOS1 showed abnormal lipid metabolism. (a, b) Immunoblot of the destabilized BLOC-1 subunits (Pallidin and Dysbindin) in lysates of liver (a) and purified primary hepatocytes (b) from loxp and cKO mice. The loxp heterozygous mice in (b) were used as another control. Note that the weak band of Pallidin in (a) indicates the protein from other non-hepatocytes in the liver. (c) Representative images showing the lipid droplets in liver frozen sections of loxp and cKO mice under indicated conditions. For mice after fasting, both Oil Red O staining and

*Figure 1 continued on next page*

*Figure 1 continued*

HE staining results are displayed. Magnified insets of boxed areas are placed on the top right corners of each picture. Scale bars, 50 μm. (**d, e**) Average area of lipid droplets in 20 random 50 μm × 50 μm square regions before (**d**) and after (**e**) starvation showing reduced lipid droplet content in cKO mice. Quantifications were performed on Oil Red O stained sections. Mean ± SEM. Two-tailed t test, \*\*\*p<0.001. (**f**) Coomassie brilliant blue staining (CBB staining) of plasma proteins in loxp and cKO mice after separation by SDS-PAGE (see the full image in *Figure 1—source data 1*). (**g**) Immunoblot of ApoE in plasma of different loxp and cKO mice, albumin is a loading control. (**h**) Lipoproteins prestained by Sudan Black B in plasma of loxp and cKO mice were separated by 4–15% gradient native PAGE at different loading volume of plasma. (**i**) Immunoblot of ApoE in lipoproteins separated by native PAGE and a second dimensional SDS-PAGE. Gel slices of prestained lipoproteins and CBB-stained proteins were used to determine the location of different lipoproteins in immunoblots. SBB, Sudan Black B. See also *Figure 1—figure supplement 1*.

The online version of this article includes the following source data and figure supplement(s) for figure 1:

**Source data 1.** Full image of CBB staining showing the comparison of control and cKO mice plasma proteins.

**Figure supplement 1.** Generation of cKO mice and the abnormal plasma lipoprotein level in cKO mice.

---

immunoblotting to the gel slices of target lanes containing separated lipoproteins. Indeed, we observed that there was more ApoE in the HDL slice of cKO mice (*Figure 1i*).

Collectively, these results suggest that cKO mice have abnormal lipid metabolism, mainly characterized by reduced lipid droplets in the liver, and altered plasma lipoprotein compositions with increase of LDL, reduction of VLDL, and increase of ApoE in HDL.

## BLOS1 deficiency reduces membrane LDLR in hepatocytes

Despite the reduced VLDL content, a common constituent of LDL and VLDL, ApoB, was significantly increased in cKO mouse plasma (*Figure 2—figure supplement 1a*), suggesting that increase of LDL in cKO mouse plasma was likely the main effect of BLOS1 deficiency on lipoproteins. VLDL is largely converted to LDL in plasma. The reduced VLDL in cKO mouse plasma is unlikely the major cause of increased LDL. Liver is the primary site for plasma LDL clearance, the accumulation of LDL in cKO mouse plasma suggested that LDL clearance by cKO mouse liver could be impaired. To test this hypothesis, we examined the endocytosis of LDL in primary hepatocytes using purified mouse LDL labeled with DiI (LDL-DiI). We observed that LDL endocytosis in cKO hepatocytes was reduced at the initial stage (*Figure 2a,b*), suggesting that less LDL binds to the cell membrane of cKO hepatocytes. We then detected the LDLR protein level by immunoblotting. We found a significant reduction of LDLR in cKO mouse liver (*Figure 2c,f* and *Figure 2—figure supplement 1b*). Immunofluorescence staining of LDLR in primary hepatocytes isolated from both control and cKO mice also showed significant reduction of LDLR in cKO mice (*Figure 2—figure supplement 1c*).

The reduction of LDLR in cKO mouse liver was not attributable to the dysfunction of either the BLOC-1 or BORC complexes, as both the *pa* mice (lack of Pallidin subunit in BLOC-1) and *Kxd1*-KO mice (lack of KXD1 subunit in BORC) showed no obvious changes in liver LDLR levels (*Figure 2d–f*). This suggests that the effect on LDLR is specific for BLOS1 alone. Considering that the uptake of LDL by other tissues (except the liver) is unaffected in cKO mice, the increase of plasma LDL is mainly attributable to impaired liver clearance of LDL.

The reduction of LDLR caused by BLOS1 deficiency was further confirmed in Hep G2 cells with stable knockdown of *BLOC1S1* (*BLOC1S1*-KD cells) (*Figure 2g*). The potential down-regulation effect of BLOS1 deficiency on LDLR transcriptional levels was excluded by qRT-PCR in two different sets of primers (*Figure 2h*) (actually, the mRNA level of LDLR was slightly increased in *BLOC1S1*-KD cells), suggesting a post-transcriptional effect on LDLR. In addition, reduction of LDLR level (as well as TfR level) in *BLOC1S1*-KD cells was recovered after the inhibition of lysosomal degradation by leupeptin (*Figure 2i*), suggesting excessive degradation of LDLR or TfR due to the loss of BLOS1. We also noted that the content of PCSK9, a known negative regulator of LDLR degradation, was not affected in cKO mouse livers (*Figure 2—figure supplement 1b*).

We next investigated whether BLOS1 could physically interact with LDLR. LDLR co-immunoprecipitated with FLAG-tagged BLOS1 (*Figure 2—figure supplement 1d*) and GST-BLOS1 fusion protein could pull-down LDLR from liver tissue lysate (*Figure 2—figure supplement 1e*), indicating an interaction between BLOS1 and LDLR. We further constructed different truncations of BLOS1, and then performed the GST pull-down experiments using mouse liver lysates. The interacting region on BLOS1 was narrowed down to residues 76–100 (*Figure 2—figure supplement 1f–h and j*).

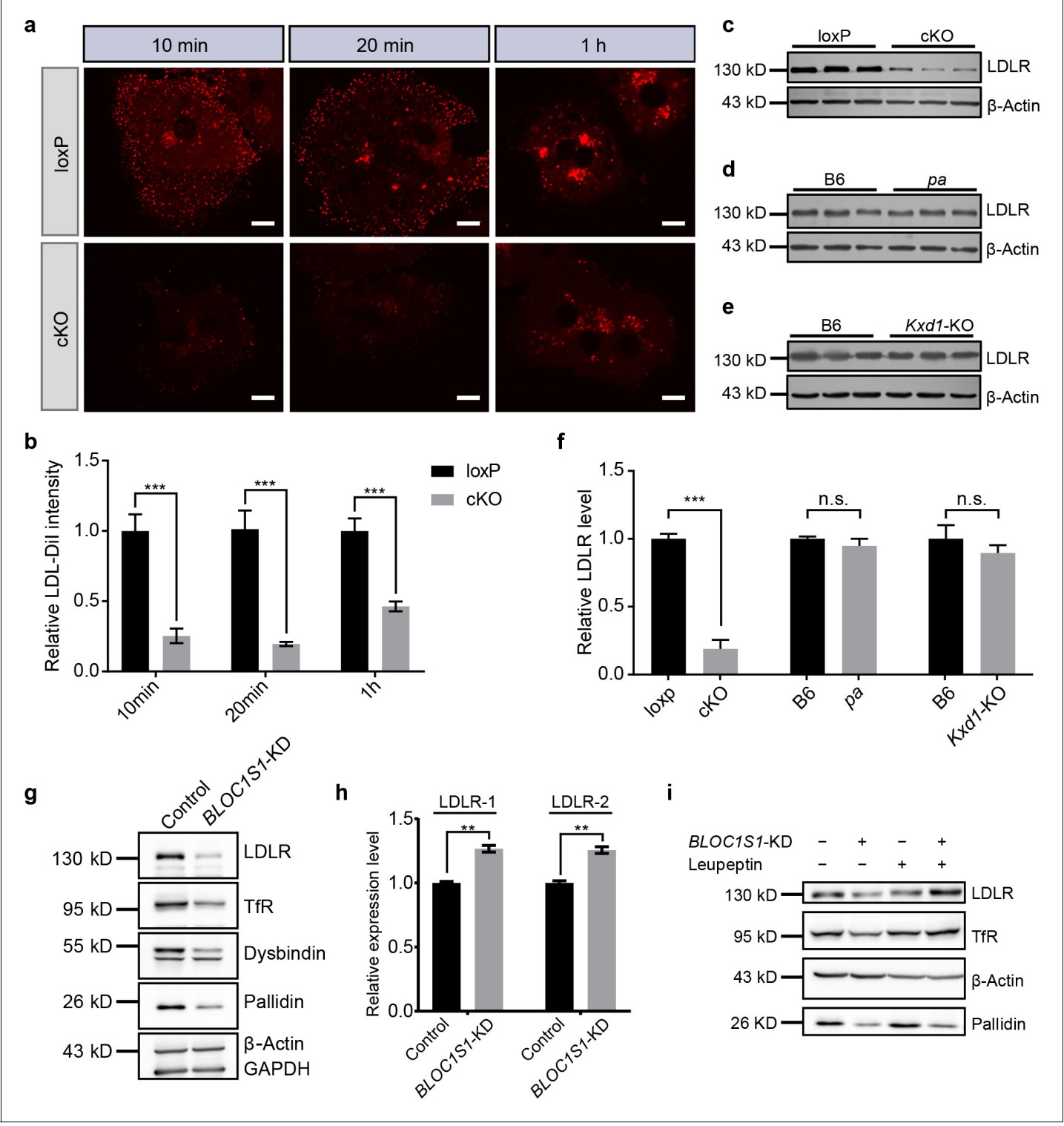

**Figure 2.** BLOS1 regulates LDLR membrane trafficking and interacts with LDLR. (**a**) Endocytosis of DiI-labeled LDL in primary hepatocytes from loxp and cKO mice after indicated time points. Scale bars, 10 μm. (**b**) Average intensity of LDL-DiI signal after endocytosis in loxp (n = 10 [10min], n = 10 [20 min], n = 11 [2h]) and cKO (n = 8 [10min], n = 9 [20 min], n = 8 [2h]) hepatocytes at indicated time points. Mean ± SEM. Two-tailed t test, ***p<0.001. (**c–e**) Immunoblot of LDLR in cKO, *pa* and *Kxd1*-KO mouse livers showing the decreased LDLR in cKO mice, but not in *pa* or *Kxd1*-KO mice. (**f**) Quantification of relative LDLR protein level (normalized to β-Actin) in *pa* (n = 3), *Kxd1*-KO (n = 6) and cKO (n = 5) mice and their control goups. Mean ± SEM. Two-tailed t test; n.s., not significant; ***p<0.001. (**g**) Detection of LDLR and TfR in *BLOC1S1* stable knockdown Hep G2 cells (*BLOC1S1*-KD cells). Destabilization of Pallidin or Dysbindin is also shown. (**h**) qRT-PCR of LDLR normalized to *GAPDH* using two different primer pairs (LDLR-1

*Figure 2 continued on next page*

*Figure 2 continued*

and LDLR-2) in control and *BLOC1S1*-KD cells. Mean ± SEM, three repeats. Two-tailed t test, **p<0.01. (i) Recovery of LDLR level (as well as TfR level) in *BLOC1S1*-KD cells after the inhibition of lysosomal degradation by treating with leupeptin. See also *Figure 2—figure supplement 1*.

The online version of this article includes the following figure supplement(s) for figure 2:

**Figure supplement 1.** Reduced LDLR in cKO mice and LDLR interacting region mapping on BLOS1.

Furthermore, the cytosolic domain of LDLR at its C-terminus could pull down BLOS1 (*Figure 2—figure supplement 1i,j*).

Taken together, these results showed that loss of BLOS1 in liver could lead to impaired LDL clearance from plasma which is caused by excessive lysosomal degradation of LDLR, and residues 76–100 on BLOS1 interacts with the cytosolic domain of LDLR at the C-terminus.

## BLOS1 localizes to microtubules and interacts with kinesin motors

To further explore the underlying mechanism of how BLOS1 regulates LDLR level, we first performed immunocytochemistry (ICC) to observe the subcellular distribution of BLOS1. In Hep G2 cells, over-expressed BLOS1 with different tags all showed puncta distribution patterns in the cytosol (*Figure 3a*). When fused to a large tag such as GFP, BLOS1-GFP mostly formed aggregates (*Figure 3a*, left). However, when fused with small tags such as Myc, FLAG or HA, BLOS1 localized to smaller puncta structures distributed more evenly in the cytosol (*Figure 3a*, right). We used small tagged BLOS1 in the following immunostaining assays in Hep G2 cells. We co-stained BLOS1 over-expressing cells with different organelle markers and found that BLOS1 partially colocalized with lysosome/MVB marker CD63 (*Figure 3c*), and almost no colocalization of BLOS1 with mitochondria (labeled by Cytochrome C) was observed (*Figure 3b*). In addition, a small fraction of BLOS1 colocalized with the early endosome marker EEA1 (*Figure 3d*), and more BLOS1 puncta colocalized with TfR labeled recycling endosomes (REs) (*Figure 3e*) or LDLR-positive vesicles (*Figure 3f*, see also *Figure 3g* for quantification), indicating a potential role of BLOS1 in the endocytic and recycling pathway.

Interestingly, when overexpressed in mouse primary hepatocytes, BLOS1 showed a tubular distribution in addition to the scattered puncta (*Figure 3h*). This observation promoted us to examine the relationship between these tubular structure and cytoskeletons. We found that these tubular structures colocalized well with α-tubulin labeled microtubules (*Figure 3i*, bottom) but not with actin filaments (labeled with phalloidin) (*Figure 3i*, top). The microtubule-localized pattern of BLOS1 in primary hepatocytes suggests that BLOS1 may be associated with microtubules.

Since BLOS1 has no reported or predicted microtubule-binding motif, we then explored whether other microtubule binding proteins were involved in the microtubular distribution of BLOS1 in primary hepatocytes. Three motor proteins (KIF3B, KIF13A, and KIF16B) from the kinesin superfamily, which have been reported to function in the anterograde transport of vesicles in endocytic system, were taken as candidates. Among these three KIF proteins, KIF3B belongs to the kinesin-2 family, while KIF13A and KIF16B are classified in the kinesin-3 family. The co-immunoprecipitation (co-IP) assays revealed that BLOS1 interacted with both KIF3B and KIF13A, but not KIF16B (*Figure 3j*). In agreement with these results, we found that the tubular structures that BLOS1 labeled were KIF13A-positive (*Figure 3k*). Together, these results suggest that BLOS1 may play a role in endosomal trafficking through interacting with KIF proteins on the microtubules.

## KIF13A transports recycling endosome-resident LDLR

We noticed that LDLR mainly localized peripherally underneath the cell membrane in primary hepatocytes (*Figure 4a*, left), while in Hep G2 cells, LDLR showed a scattered distribution (*Figure 4a*, right). We further explored the different distribution pattern of LDLR in Hep G2 cells and mouse primary hepatocytes. We found that the peripherally distributed LDLR in primary hepatocytes were mostly located on clathrin-coated vesicles (labeled by clathrin light chain A, Clta) (*Figure 4—figure supplement 1a*), while only a portion of LDLR were colocalized with CLTA in Hep G2 cells (*Figure 4—figure supplement 1b*), suggesting that LDLR in primary hepatocytes were recycled faster to the cell membrane to achieve their physiological role to recycle the majority of LDLR in plasma. In addition, LDLR partially colocalized with other endocytic compartments, such as early endosomes

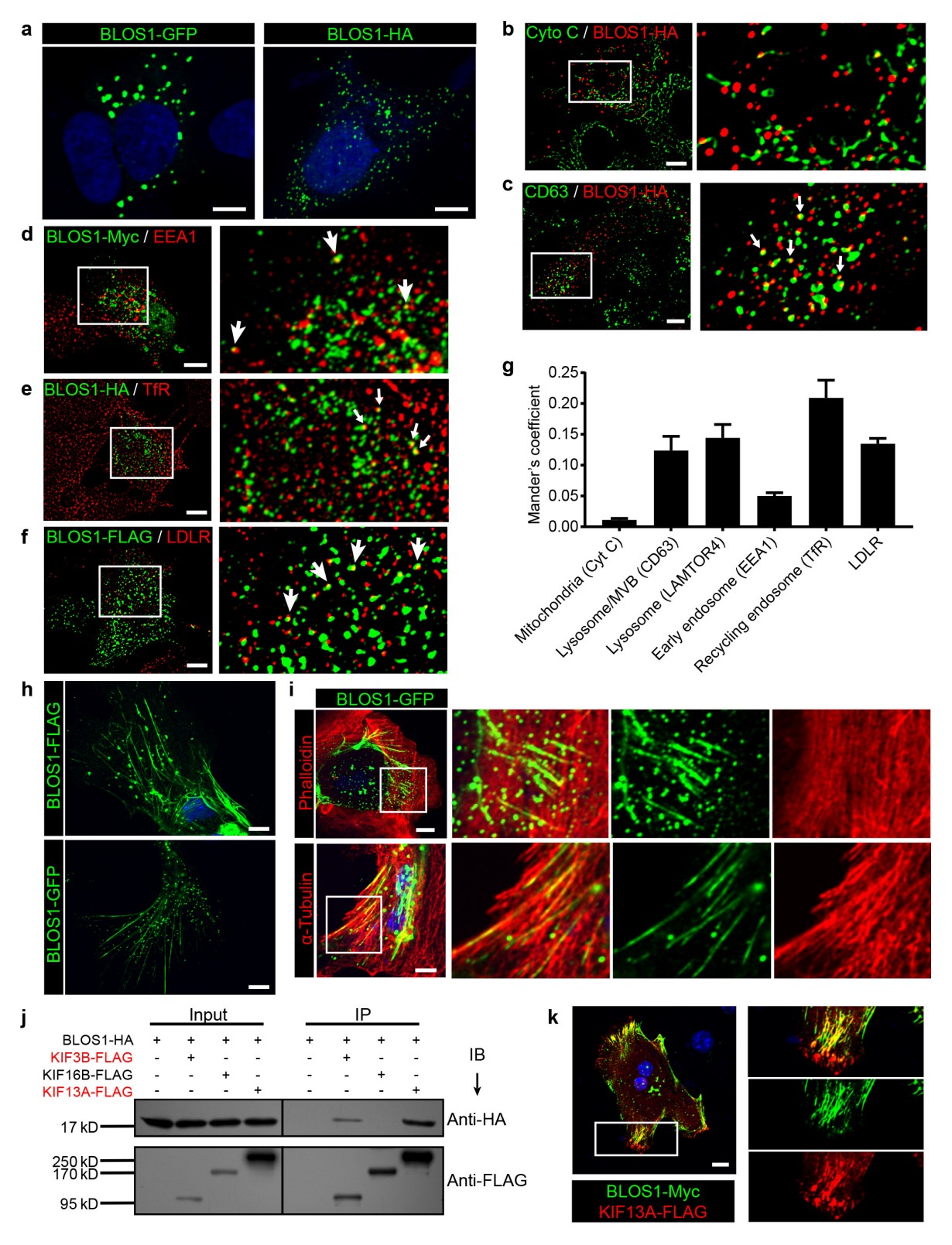

**Figure 3.** BLOS1 localizes to microtubules and interacts with kinesin motors. (**a**) Representative confocal images of puncta patterns of BLOS1-GFP-C2 and immunofluorescence stained BLOS1-HA in Hep G2 cells. (**b–f**) BLOS1 partially colocalizes with multivesicular body/lysosome marker CD63 (**c**, white arrows), early endosome marker EEA1 (**d**), recycling endosome marker TfR (**e**) and LDLR vesicles (**f**) in Hep G2 cells, while almost no colocalization was observed between BLOS1-HA and mitochondria marker Cytochrome C (**b**). Magnified insets of boxed areas are shown on right. (**g**) Quantification of

*Figure 3 continued*

Mander's colocalization coefficient showing the percentage of BLOS1 that colocalizes with other proteins or organelle markers. n = 5, 6, 6, 5, 6 and 6 from left to right, respectively. Data are presented as Mean ± SEM. (h) Representative confocal images showing tubular structures of overexpressed BLOS1 with different tags in mouse primary hepatocytes. (i) Tubular BLOS1-GFP expressed in mouse primary hepatocytes distributes on microtubule (indicated by α-Tubulin) (bottom), but not actin filaments (labeled by Phalloidin, top). Merged and single labeling images of magnified insets of boxed areas are shown in bottom panels of each figure. (j) co-IP (immunoprecipitation) of BLOS1-HA with co-overexpressed KIF3B-FLAG or KIF13A-FLAG, but not KIF16B-FLAG, in HEK293T whole cell lysate after incubation with anti-FLAG beads, followed by immunoblotting (IB). (k) BLOS1 tubules colocalize well with KIF13A-FLAG positive microtubules in mouse primary hepatocytes. Merged and single labeling images of magnified insets of boxed areas are shown on right. Scale bars in all pictures, 10 μm.

(*Figure 4—figure supplement 1c*) and recycling endosomes (*Figure 4—figure supplement 1d*) in Hep G2 cells. Therefore, Hep G2 cells were more suitable than mouse primary hepatocytes for investigating the recycling of LDLR via endocytic compartments.

Expression of KIF13A-GFP in primary hepatocytes showed the majority of LDLR were enriched at some peripheral domains (*Figure 4b*), while the KIF13A truncation lacking the motor region (KIF13A-ST) retained LDLR on dispersed vesicles in the cytosol (*Figure 4c*), suggesting that KIF13A participates in the transport of LDLR to the cell periphery, and dysfunction of KIF13A disturbs the normal trafficking of LDLR.

As BLOS1 is seen on endosomal structures in Hep G2 cells (*Figure 3*), LDLR was localized to the cell periphery in full-length KIF13A-FLAG overexpressed Hep G2 cells, and these LDLR signals colocalized with the RE marker TfR (*Figure 4d* and *Figure 4—source data 1*), suggesting that KIF13A functions on REs during LDLR recycling. Besides, the rigor mutant of KIF13A (KIF13A-R), an ATPase activity-lacking point mutant which binds to microtubule but is unable to move along the microtubule (*Guardia et al., 2016*; *Nakata and Hirokawa, 1995*), anchored LDLR and REs on KIF13A-R positive microtubules (*Figure 4e*, *Figure 4—figure supplement 2a* and *Figure 4—video 1*). This suggests a critical role of KIF13A in RE-dependent LDLR transport. As a negative control, neither full-length KIF13A nor KIF13A-R mutant affects the cellular distribution of lysosome/MVB marker CD63 (*Figure 4—figure supplement 2b,c* and *Figure 4—video 2*).

In agreement with a previous report that KIF13A functions in RE tubule morphogenesis (*Delevoye et al., 2014*), overexpressed full-length KIF13A elongated tubular REs (labeled by EHD3), while KIF13A-ST truncation had no this effect (*Figure 4f,g*). We noticed that the majority of actively moving Rab11A-positive recycling endosomes were also labeled by LDLR in live-cell imaging (*Figure 4—video 3* and *Figure 4—figure supplement 1d*). Live-cell imaging revealed that, in contrast to the re-distribution of REs (labeled by RAB11A) to the cell periphery by KIF13A (*Figure 4—video 4*), expression of KIF13A-R affects the movement of REs (labeled by RAB11A), most REs distributed along the KIF13A-R positive microtubules and were almost static during the 2 min imaging period (*Figure 4h* and *Figure 4—figure supplement 2d* and *Figure 4—video 1*), indicating that the LDLR anchored on microtubules through KIF13A-R on REs.

Furthermore, we found that KIF13A interacted with LDLR in the co-IP assays (*Figure 4i*), and knockdown of *KIF13A* resembles the reduction of LDLR and TfR occurred in BLOS1 deficiency (*Figure 4j*). From these results, we firstly report that LDLR in REs is a cargo transported by KIF13A, and dysfunction of KIF13A affects the cellular distribution and homeostasis of LDLR on plasma membrane.

## BLOS1 acts as an adaptor protein of KIF3 complex in the regulation of RE trafficking

In both Hep G2 cell lines (*Figure 5a*) and mouse primary hepatocytes (*Figure 5—source data 1*), KIF3A or KIF3B was evenly distributed in cells without any enrichment of LDLR in specific regions or puncta. Similarly, The expression of KIF3B rigor mutant (KIF3B-R) did not affect the cellular localization of LDLR on REs (labeled by TfR) (*Figure 5b*). However, by live-cell imaging analysis, we found that, unlike the active movement in control cells, most of the RE puncta in KIF3B-R expressing cells were anchored at dispersed sites on KIF3B-R-positive microtubules (*Figure 5c* and *Figure 5—video 1*), suggesting that KIF3B may play a role in the trafficking of REs. In our observations, non-specific blocking of all microtubule-based vesicle transport by KIF3B-R were excluded as we can observe the transport of lysosomes (labeled by dextran-TMR) using live-cell imaging (*Figure 5—video 2*).

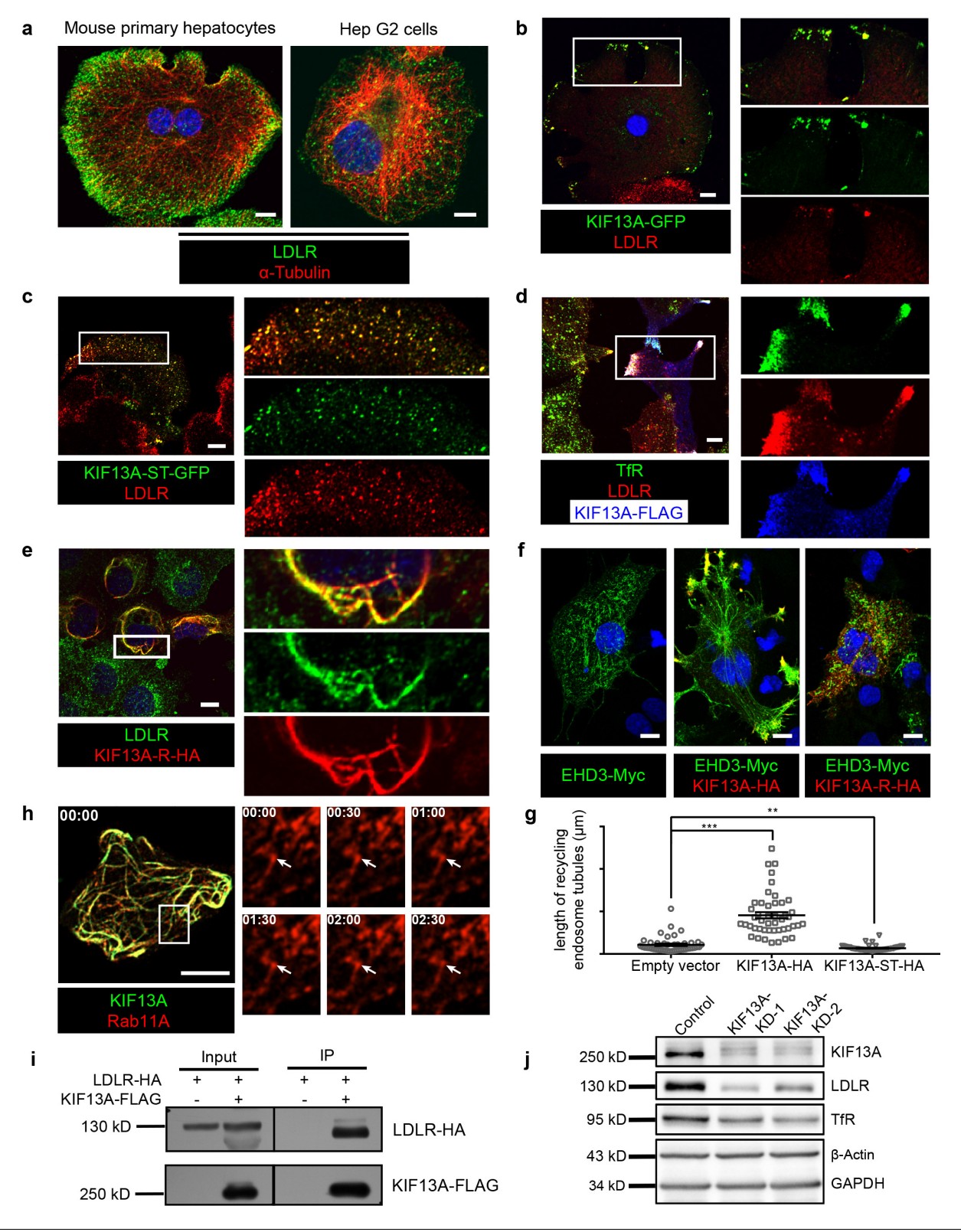

**Figure 4.** KIF13A transports recycling endosome-resident LDLR. (a) Representative confocal images showing different distribution patterns of endogenous LDLR (green) in mouse primary hepatocytes and Hep G2 cells. Microtubules were co-labeled with α-tubulin (red) to show the cell morphology. (b) Immunofluorescence staining of LDLR in mouse primary hepatocytes after the expression of KIF13A-GFP. Merged and single labeling images of magnified insets of boxed areas are shown on right. (c) Retention of LDLR in the cytosol of mouse primary hepatocytes by expressing

*Figure 4 continued on next page*

*Figure 4 continued*

KIF13A-ST truncation. Merged and single labeling images of magnified insets of boxed area are shown on right. (d) Peripherally distributed LDLR driven by KIF13A-FLAG colocalizes with TfR labeled recycling endosomes in Hep G2 cells. Merged and single labeling images of magnified insets of boxed areas are shown on right. (e) Stuck of LDLR on microtubules caused by KIF13A-R point mutant expression in Hep G2 cells. Merged and single labeling images of magnified insets of boxed area are shown on right. (f) Representative immunostaining results showing the elongation of recycling endosome tubules caused by full-length KIF13A and shortening of recycling endosome tubules caused by KIF13A-ST truncation in Hep G2 cells. (g) Average tubular recycling endosomes length measured in empty vector (n = 44), KIF13A-HA (n = 46) and KIF13A-ST-HA (n = 46) expressing cells in (f). Mean ± SEM, two-tailed t test, **p<0.01, ***p<0.001. (h) Confocal live-cell microscopy of RAB11A-Scarlet labeled recycling endosomes in KIF13A-R-GFP expressing Hep G2 cells. Magnified insets (of boxed area) of consecutive time-lapse images (image/30 s) showed that most of the recycling endosomes were retained on KIF13A-R positive microtubules (see *Figure 4—video 1*). White arrows indicate representative recycling endosomes that were almost static during the imaging period. Time stamps are in the format of minutes: seconds. (i) Co-IP of LDLR-HA with co-overexpressed KIF13A-FLAG in HEK293T whole cell lysate after incubation with anti-FLAG beads, followed by immunoblotting. (j) Immunoblots of LDLR and TfR in *KIF13A* stable knockdown cells. Scale bars in all pictures, 10 μm. See also *Figure 4—figure supplements 1* and *2*, *Figure 4—videos 1*, *2*, *3* and *4*, *Figure 4—source data 1*.

The online version of this article includes the following video, source data, and figure supplement(s) for figure 4:

**Source data 1.** KIF13A transport LDLR to cell periphery in both control and cKO hepatocytes.

**Figure supplement 1.** LDLR locates on multiple endocytic compartments in Hep G2 cells.

**Figure supplement 2.** KIF13A and its rigor mutant do not affect lysosome distribution.

**Figure 4—video 1.** Live-cell imaging of Hep G2 cells transfected with KIF13A-R-Scarlet and RE marker RAB11A-GFP.

https://elifesciences.org/articles/58069#fig4video1

**Figure 4—video 2.** Live-cell imaging of Hep G2 cells transfected with KIF13A-R-GFP and then incubated with Dextran-TMR (molecular weight = 10,000) to label lysosomes.

https://elifesciences.org/articles/58069#fig4video2

**Figure 4—video 3.** Live-cell imaging of Hep G2 cells transfected with LDLR-GFP and Rab11a-Scarlet.

https://elifesciences.org/articles/58069#fig4video3

**Figure 4—video 4.** Live-cell imaging of Hep G2 cells transfected with KIF13A-GFP and Rab11a-Scarlet.

https://elifesciences.org/articles/58069#fig4video4

In addition, we observed that BLOS1 could be recruited to microtubules by KIF3B-R, and the rigor mutants of other two KIF proteins of kinesin-2 (KIF3A or KIF3C) both had the same effect, while KIF13A-R, which also interacted with BLOS1, or KIF5B-R (a rigor mutant of a kinesin-1 component) did not affect BLOS1 localization (*Figure 5d*). Furthermore, when KIF3A and KIF3B, which forms a heterodimer, were co-transfected with BLOS1 in Hep G2 cells, these two KIF proteins were redistributed to BLOS1 positive puncta structures (*Figure 5e*). One possible explanation for these results is that BLOS1 may form a tight protein complex with KIF3A/B or KIF3A/C heterodimers.

Due to the lack of a cargo-binding region in KIF proteins of kinesin-2, adaptor proteins are required to form a functional kinesin-2 complex for cargo transport. Therefore, we cloned a known kinesin-2 adaptor protein gene *KAP3* and found that when co-expressed with KIF3B-R or KIF3A/B heterodimer, KAP3 show similar localization pattern as BLOS1 (*Figure 5f,g*), indicating that BLOS1 may act as a new adaptor protein of kinesin-2.

To avoid a possible redundant function between KIF3B and KIF3C, we chose the common subunit KIF3A as the target to be knocked down. Immunoblotting assays revealed that KIF3A knock-down also caused decreased LDLR levels in Hep G2 cells (*Figure 5h*). Together, our data indicate that BLOS1 may be a new adaptor protein participating in the assembly of functional kinesin-2 complexes, which regulate RE trafficking at dispersed sites on microtubules.

## KIF3 is essential for the long-range anterograde transport of REs driven by KIF13A

To explore the detailed mechanism underlying the regulation of RE trafficking by kinesin-2, we labeled REs in both control and *KIF3A*-KD cells with RAB11A-GFP and then analyzed their motion using live-cell imaging. We found that in these cells, the movement of REs could be categorized into two groups, a portion of REs was static or showed slightly local motion, whereas the other REs moved quickly in anterograde or retrograde transport directions. We named these quickly moving REs as 'active REs'. Further analysis revealed that the motion of anterograde transported active REs in *KIF3A*-KD cells was significantly different from that of control cells (*Figure 6a–d*). In control cells, active REs in anterograde transport sometimes paused and then usually continued their motion in

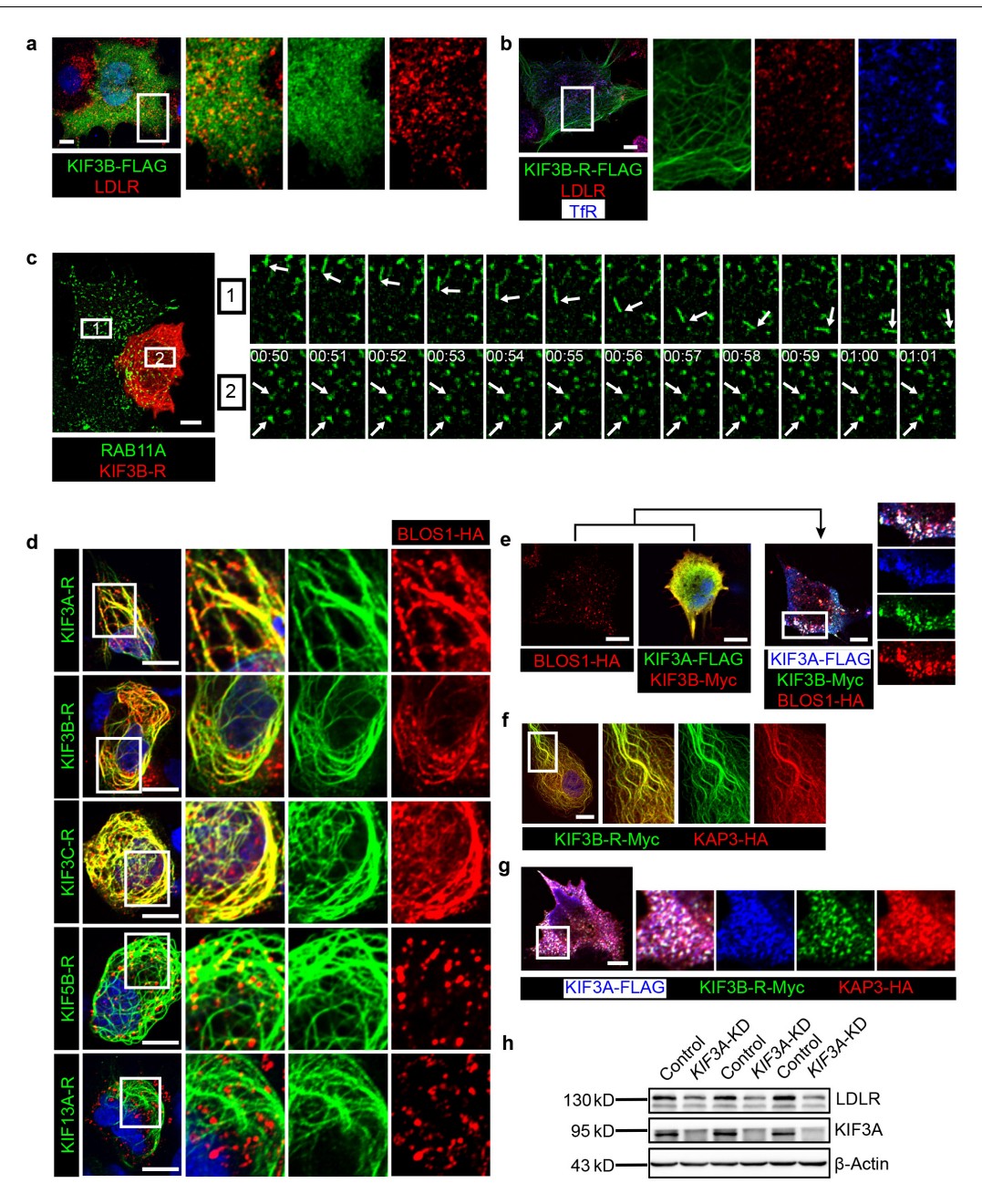

**Figure 5.** BLOS1 acts as an adaptor protein of KIF3 complex in the regulation of RE anterograde transport. (**a**) Immunofluorescence staining of LDLR in KIF3B-FLAG expressing Hep G2 cells. Merged and single labeling images of magnified insets of boxed areas are shown on right. (**b**) Puncta distribution of LDLR and TfR in KIF3B-R expressing Hep G2 cells. Merged and single labeling images of magnified insets of boxed area are shown on right. (**c**) Confocal live-cell microscopy of RAB11A-GFP labeled recycling endosomes in KIF3B-R expressing and non-transfected Hep G2 cells. Magnified insets (of box 1 and 2 in the same time points) of consecutive time-lapse images (image/1 s) showed that recycling endosomes in control cells had active motion, while this movement was impaired in KIF3B-R expressing cells (see *Figure 5—video 1*). White arrows indicate representative recycling endosomes in control and KIF3B-R-expressing cells. Time stamps are in the format of minutes: seconds. (**d**) Retention of BLOS1 on microtubules caused by co-expressed KIF3A, KIF3B and KIF3C rigor mutants, but not KIF5B or KIF13A rigor mutants in Hep G2 cells. Merged and single labeling images of magnified insets of boxed areas are shown on right. (**e**) Co-expression of BLOS1 with both KIF3A and KIF3B redistributed KIF3A/B to BLOS1-positive puncta in Hep G2 cells. Merged and single labeling images of magnified insets of boxed area are shown on right. (**f, g**) Similar retention of KAP3 on microtubules caused by KIF3B-R (**f**) and redistribution of KIF3A/B resulting from the co-expression of KAP3 (**g**) in Hep G2 cells. Merged and single labeling images of magnified insets of boxed area are shown on right. (**h**) Immunoblot of LDLR in *KIF3A* stable knockdown Hep G2 cells (KIF3A-KD cells). Scale bars in all pictures, 10 μm. See also *Figure 5—video 1* and *2*, *Figure 5—source data 1*.
The online version of this article includes the following video and source data for figure 5:

*Figure 5 continued on next page*

*Figure 5 continued*

**Source data 1.** KIF3A distributes evenly in primary hepatocytes.
**Figure 5—video 1.** Live-cell imaging of Hep G2 cells transfected with (or without) KIF3B-R-Scarlet and RE marker RAB11A-GFP.
https://elifesciences.org/articles/58069#fig5video1
**Figure 5—video 2.** Live-cell imaging of Hep G2 cells transfected with KIF3B-R-GFP and then incubated with Dextran-TMR (molecular weight = 10,000) to label lysosomes.
https://elifesciences.org/articles/58069#fig5video2

their original anterograde direction (*Figure 6a* and *Figure 6—video 1*). In contrast, after the knock-down of *KIF3A* (*Figure 6b* and *Figure 6—video 1*) or *BLOC1S1* (*Figure 6c* and *Figure 6—video 2*), active REs frequently moved backward after paused anterograde transport. Most of these REs moved back to the perinuclear endocytic recycling compartment (ERC) of their origin by following the backward movement (*Figure 6—video 1*), suggesting a reduction in the percentage of REs that could eventually be destined to the cell periphery.

The above results showed that both kinesin-2 and kinesin-3 participate in the transport of REs. We wondered whether they cooperate in this process. To test whether kinesin-2 is required for the normal function of kinesin-3, we expressed KIF13A-GFP in both control and *KIF3A*-KD cells and then analyze the behaviors of these KIF13A-positive tubular structures before and after KIF3A deficiency. We found that, unlike the successful extension to the cell periphery in control cells (*Figure 6d* and *Figure 6—video 3*), most of the tips of KIF13A-positive tubules in *KIF3A*-KD cells showed inter-rupted movement when they encountered other KIF13A tubules (*Figure 6e* and *Figure 6—video 3*). Some KIF13A tubule tips moved forward and backward repeatedly on the same track (*Figure 6f*, arrow 2), while the others paused and then moved backward on another track resembling the behav-ior of REs in KIF3A-KD cells (*Figure 6f*, arrow 1). Similar abnormal movement of KIF13A-positive tubules could be observed when the dominant-negative mutant KIF3B-R were expressed (*Figure 6—video 4*). In addition, KIF3A-KD cells showed normal microtubule architecture (*Figure 6—figure sup-plement 1a*) and dynamics (indicated by motility of EB1-GFP positive microtubule tips) (*Figure 6—figure supplement 1b* and *Figure 6—video 5*), suggesting unaffected microtubule network after KIF3A deficiency.

In line with the above results, when KIF13A-R and KIF3B-R were co-expressed, the distribution of LDLR is determined by KIF13A-R, indicating that KIF13A could function upstream of KIF3B (*Fig-ure 6—figure supplement 1c*). However, when KIF3A, KIF3B, and BLOS1 were co-overexpressed to form an intact kinesin-2 complex, peripheral accumulation of LDLR which occurred in the case of KIF13A expression, was not observed (*Figure 6—figure supplement 1d*), suggesting that kinesin-2 itself may not drive the long-range anterograde transport of REs, and KIF3-BLOS1 and KIF13A may play different roles in this process.

To further determine the distribution of KIF13A-R and KIF3B-R on microtubules, we co-stained these two rigor mutants with antibodies to acetylated and tyrosinated tubulin, as it has been shown that tyrosinated tubulin has a broader distribution than centrally located acetylated tubulin (*Guardia et al., 2016*), which resembles the distribution pattern of KIF3B-R and KIF13A-R. We found that both KIF13A-R and KIF3B-R located on acetylated microtubules (*Figure 6—figure supplement 1e,f*, top), and KIF3B-R but not KIF13A-R was additionally found on peripherally distributed tyrosi-nated microtubules (*Figure 6—figure supplement 1e,f*, bottom). These observations indicated that KIF13A and KIF3B may walk along the same set of microtubule tracks with acetylated tubulin and post-translational modifications of tubulin may contributes to the association of KIF13A and KIF3B with microtubules.

## KIF3 and BLOS1 function at specific microtubule intersections

It has been reported that cargoes (lysosomes) pause at intersection points between multiple micro-tubules and switch to another microtubule track at some specific intersections, and these microtu-bule intersections serve as important sites for direction determination during cargo transport (*Bálint et al., 2013*; *Bergman et al., 2018*; *Verdeny-Vilanova et al., 2017*). We found that, after knockdown of *KIF3A*, when encountering other KIF13A-GFP positive tubules, tips of KIF13A tubules were unable to pass through and move on (*Figure 7a,b*). As the KIF13A-positive tubules move along

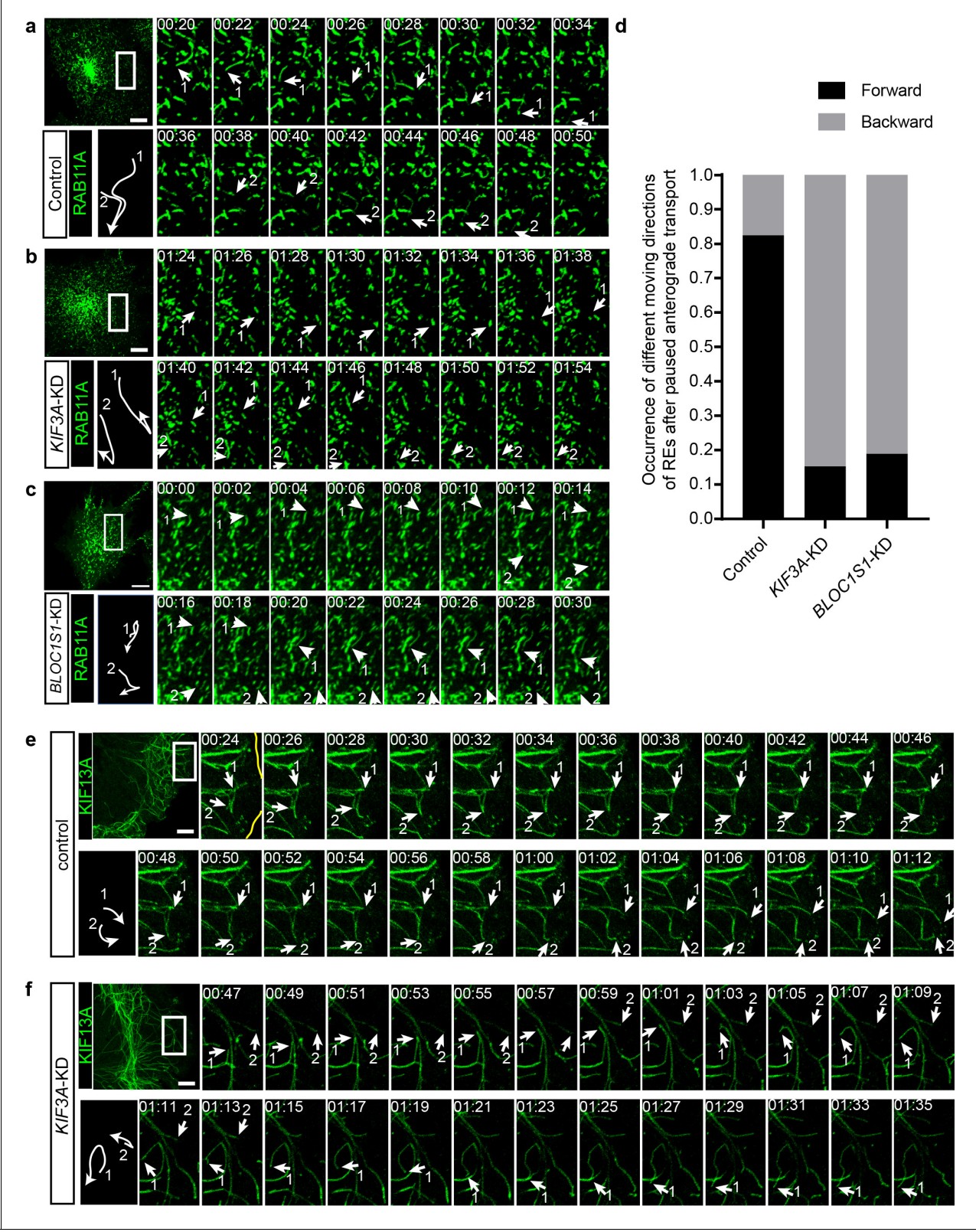

**Figure 6.** KIF3 is essential for the long-range anterograde transport of REs driven by KIF13A. (a–c) Confocal live-cell microscopy of RAB11A-GFP labeled recycling endosomes in control (a), *KIF3A*-KD (b) and *BLOC1S1*-KD (c) cells (see also *Figure 6—videos 1* and *2*). Magnified insets (of boxed areas) of consecutive time-lapse images (image/2 s) showed that, compared to control cells, the occurrence of backward movement of recycling endosomes following paused anterograde transport is significantly increased in *KIF3A*-KD and *BLOC1S1*-KD cells. White arrows labeled 1 and 2

*Figure 6 continued on next page*

*Figure 6 continued*

indicate representative recycling endosomes in these cells. The arrows were drawn perpendicularly to the direction of motion and were reversed when moving in the opposite direction. A trajectory diagram of the representative recycling endosomes is shown on the left bottom. Time stamps are in the format of minutes: seconds. (d) Histogram showing the occurrence of forward and backward motion of recycling endosomes following paused anterograde transport in control (n = 50) and *KIF3A*-KD (n = 124) cells. (e, f) Confocal live-cell microscopy of KIF13A-GFP tubules in control and *KIF3A*-KD cells (see *Figure 6—video 3*). Magnified insets (of boxed areas) of consecutive time-lapse images (image/2 s) showed impaired long-range transport of KIF13A tubules in *KIF3A*-KD cells. White arrows labeled 1 and 2 indicate tips of representative KIF13A-positive tubules in control (e) and *KIF3A*-KD (f) cells. The arrows were drawn perpendicularly to the direction of motion and were reversed when moving in the opposite direction. A trajectory diagram of the representative KIF13A tubule tips is shown on the left bottom. Time stamps are in the format of min:s. Scale bars in all pictures, 10 μm. See also *Figure 6—figure supplement 1*, *Figure 6—video 1, 2, 3, 4* and *5*.

The online version of this article includes the following video and figure supplement(s) for figure 6:

**Figure supplement 1.** KIF3 and KIF13A bind to the same set of microtubule tracks with acetylation tubulin.

**Figure 6—video 1.** Live-cell imaging of REs in control and *KIF3A*-KD cells.

https://elifesciences.org/articles/58069#fig6video1

**Figure 6—video 2.** Live-cell imaging of REs in *BLOC1S1*-KD cells.

https://elifesciences.org/articles/58069#fig6video2

**Figure 6—video 3.** Live-cell imaging of KIF13A tubules in control and *KIF3A*-KD cells.

https://elifesciences.org/articles/58069#fig6video3

**Figure 6—video 4.** Live-cell imaging of Hep G2 cells transfected with KIF13A-GFP and KIF3B-R-Scarlet.

https://elifesciences.org/articles/58069#fig6video4

**Figure 6—video 5.** Live-cell imaging of microtubule dynamics in control (left) and *KIF3A*-KD (right) cells.

https://elifesciences.org/articles/58069#fig6video5

microtubules, the site where a tip of KIF13A tubule encounters another KIF13A tubule may represent the intersection point between these two microtubules (*Figure 7a*). This raised the possibility that the BLOS1-dependent kinesin-2 complex may function at specific microtubule intersections.

To test this idea, we first detected the distribution of BLOS1 on microtubules using super-resolution microscopy and found that a certain percentage of BLOS1 puncta (77.6 ± 4.6%, n = 8 cells, mean ± SD) located at the intersections of microtubules in Hep G2 cells (*Figure 7c*, see also *Figure 7—video 1* for z-stack reconstruction movie), which could also be observed in mouse primary hepatocytes (*Figure 7—figure supplement 1a,b*), indicating that kinesin-2 may function at these sites. Then we observed the behavior of recycling endosomes (indicated by RAB11A-GFP) on microtubule tracks (stained by Tubulin Tracker Deep Red) in live cells using Zeiss LSM 880 system's Airyscan super-resolution module. We found that a large proportion of pausing events of recycling endosomes occurred at microtubule intersection sites, and in most cases, recycling endosomes moved on without changing their motion polarity after the pausing (*Figure 7—video 2*). But under certain conditions, recycling endosomes reversed their moving direction after their pausing at specific microtubule intersections (*Figure 7—video 3*). Furthermore, statistic results showed that the frequency of reversed movement of recycling endosomes after paused anterograde transport at specific microtubule intersections was increased by knockdown of either *BLOC1S1* (77.3 ± 8.4%) or *KIF3A* (78.7 ± 6.6%) as compared to control cells (10.1 ± 1.8%) (*Figure 7d*), suggesting that dysfunction of BLOS1-dependent kinesin-2 complex alters the behavior of recycling endosomes at specific microtubule intersections by changing their motion polarity and finally reduces the recycling of LDLR.

In summary, our observations suggest that BLOS1 acts as an adaptor for kinesin-2 to assist the LDLR cargo transportation driven by kinesin-3 at specific microtubule-microtubule intersections where hurdles may preclude the cargo for smooth long-range anterograde transport (*Figure 8*). Our results reveal a novel function of BLOS1 in mediating a kinesin switch at microtubule intersections.

## Discussion

Our results support a model in which kinesin-2 motor KIF3 (and its adaptor protein BLOS1) functions at checkpoints on microtubules during anterograde transport of REs driven by kinesin-3 motor KIF13A (*Figure 8*). The role of BLOS1 in this process is considered to be independent of BLOC-1 or BORC for the following reasons: (1) neither BLOC-1-deficient *pa* mice nor BORC-deficient *Kxd1*-KO

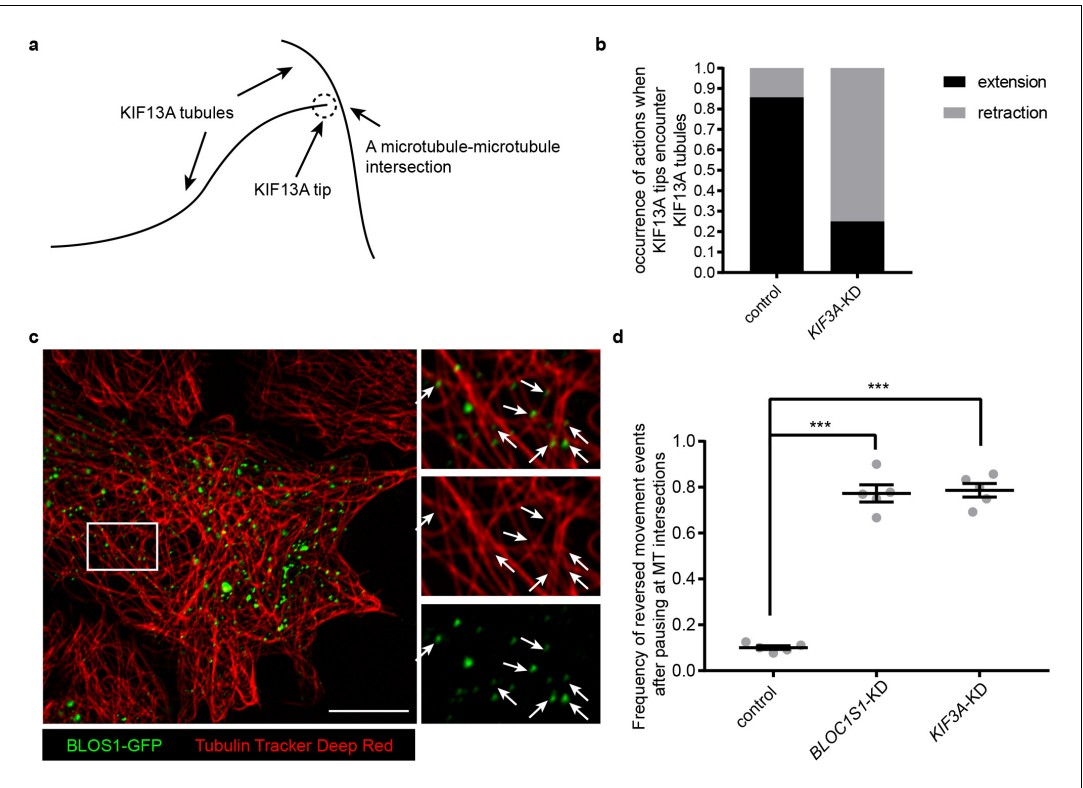

**Figure 7.** KIF3 and BLOS1 function at specific microtubule intersections. (**a**) Schematic of tip of one KIF13A-GFP tubule encounters another KIF13A-GFP tubule. (**b**) Histogram showing the occurrence of extension and retraction action when tips of KIF13A tubule encounter other KIF13A tubules in control (n = 64) and *KIF3A*-KD (n = 100) cells. (**c**) Super-resolution microscopy of the BLOS1-GFP puncta (green) on microtubule tracks (red, labeled by Tubulin Tracker Deep Red) in Hep G2 cells. Merged and single labeling images of magnified insets of boxed area are shown on right. Arrows indicate BLOS1 puncta that locate near microtubule intersections. (**d**) Frequency of reversed movement after pausing at specific microtubule intersection was increased in both *BLOC1S1*-KD and *KIF3A*-KD cells as compared to control cells. Data are presented as Mean ± SEM, n = 5 cells. Two-tailed t test, \*\*\*p<0.001. Scale bars in all pictures, 10 μm. See also *Figure 7—figure supplement 1*, *Figure 7—videos 1*, *2* and *3*.

The online version of this article includes the following video and figure supplement(s) for figure 7:

**Figure supplement 1.** BLOS1 also showed puncta pattern in primary hepatocytes and locates near microtubule intersections.
**Figure 7—video 1.** 3D-reconstruction of super-resolution Z-stack images of BLOS1-HA (green) and alpha-tubulin (red).
https://elifesciences.org/articles/58069#fig7video1
**Figure 7—video 2.** Live-cell imaging of recycling endosomes (indicated by RAB11A-GFP) moving on microtubule tracks (stained by Tubulin Tracker Deep Red probe) showing the passage of recycling endosomes after pausing at specific microtubule intersections.
https://elifesciences.org/articles/58069#fig7video2
**Figure 7—video 3.** Live-cell imaging of recycling endosomes (indicated by RAB11A-GFP) moving on microtubule tracks (stained by Tubulin Tracker Deep Red probe) showing the reversed movement of recycling endosomes at specific microtubule intersections.
https://elifesciences.org/articles/58069#fig7video3

mice showed reduced LDLR level; (2) *KIF3A*-KD cells resembled BLOS1-deficient phenotype both in LDLR level and abnormal movement of REs, indicating a kinesin-2-related function. But it remains possible that BLOS1 deficiency caused by dysfunction of BORC may further enhance LDLR degradation due to the proximity of ERC and perinuclearly clustered lysosomes since it has been reported that clustering LEs/lysosomes at the MTOC would generally enhance aggregate degradation and macroautophagy (*Bae et al., 2019*; *Korolchuk et al., 2011*).

As for the complexity in kinesin motors and cargoes, it has been reported that one kinesin could use different adaptor proteins for various cargoes and different kinesins may transport one cargo on distinct microtubule tracks (*Guardia et al., 2016*; *Hirokawa et al., 2009*). We found here that KIF3 functions downstream of KIF13A in the same trafficking pathway of REs. This observation expands knowledge of how cargoes are transported by kinesins. We suspected that BLOS1 plays a key role in the cargo switch between KIF3 and KIF13A. First, BLOS1 could interact with both KIF3 and KIF13A,

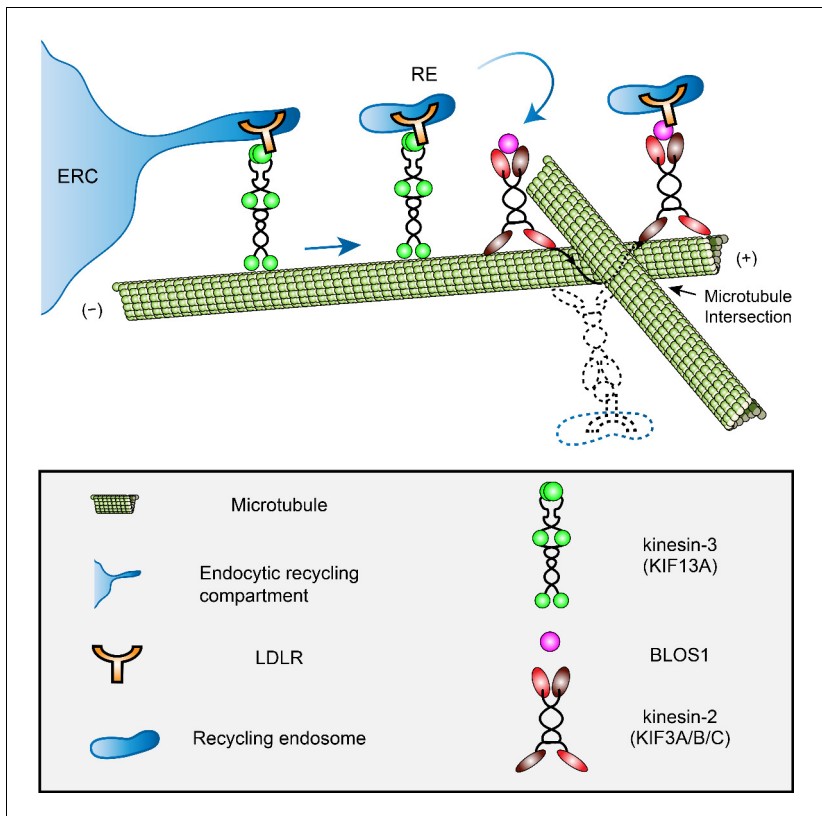

**Figure 8.** A proposed model of how BLOS1 coordinates kinesin-3 and kinesin-2 at microtubule intersections. KIF13A drives the generation and extension of tubules at the initial peripheral transport of recycling endosomes (REs) from the endocytic recycling compartment (ERC). When REs (and associated LDLR) arrive at specific microtubule-microtubule intersections, their motion will be paused by steric hindrance. At these sites, through the interaction with kinesin-3 (KIF13A) and LDLR, BLOS1 (and kinesin-2 heterodimers) assists the REs in overcoming obstacles probably by switching between microtubule protofilaments and finally causing the rotation of cargo along the microtubule surface. Dysfunction of KIF3 or BLOS1 results in the inability of REs to pass through these sites and frequently backward movements at specific microtubule intersections. Thus, impaired recycling causes alternative lysosomal degradation of LDLR. See also *Figure 8—videos 1* and *2*.

The online version of this article includes the following video(s) for figure 8:

**Figure 8—video 1.** Live-cell imaging of BLOS1-GFP puncta showing the frequent z-position change of BLOS1 puncta during recording period.

https://elifesciences.org/articles/58069#fig8video1

**Figure 8—video 2.** Live-cell imaging of BLOS1-GFP in Hep G2 cells stained with Tubulin-Tracker (Thermo Fisher).

https://elifesciences.org/articles/58069#fig8video2

and thus acts as a bridge between these two motor proteins. Second, considering the fact that KIF3 lacks cargo-binding domain, BLOS1 could act as an adaptor for a cargo protein and then switch it with KIF13A through their interaction. Thus, we thought that the interaction between BLOS1 and KIF13A is important and necessary for the cargo switch between kinesins for passing through obstacles.

Previous *in vivo* and *in vitro* studies have shown that microtubule intersections may serve as the pausing site for cargoes transport by kinesins when the size of cargoes are comparable to intersection spacing (*Bálint et al., 2013*; *Verdeny-Vilanova et al., 2017*), and it has been suggested that both the intersection geometry and motor number influence cargo routing at these intersections (*Bergman et al., 2018*; *Erickson et al., 2013*). In addition, members of the heterodimeric kinesin-2 subfamily switch protofilaments during their processive movements, which may contribute to the overcoming of microtubule obstacles (*Brunnbauer et al., 2012*; *Hoeprich et al., 2017*). Such off-axis mode motion of cargoes were also observed at microtubule intersections in vivo (*Verdeny-*

*Vilanova et al., 2017*). Therefore, in the context of recycling endosome trafficking, KIF3 may be used to pass through these intersections by protofilament switching. In agreement with this hypothesis, we observed that BLOS1 puncta showed similar off-axis mode motion during the time period of live-cell imaging (*Figure 8—videos 1* and *2*). Although we found that KIF3 (and BLOS1) is essential for the circumvention of these intersections after paused anterograde transport of recycling endosomes, the detailed mechanism needs further investigation. It may involve other microtubule-associated proteins and the geometry changes of microtubule intersections. Three possible choices of cargo transport at the microtubule intersections are: 1) continuing movement after bypassing the hurdles, 2) switching to other microtubules by changing directions or destinations, 3) moving backward to the original site. Although we here provide one example, it remains intriguing how the kinesins coordinate in these movements.

We recently reported the lipidomic profiling of *pa* mice (another BLOC-1 subunit mutant) (*Ma et al., 2019*) by showing an anti-anthrogenic effect. Reduced lipid droplets in hepatocytes are seen in both *pa* and *Bloc1s1*-cKO mice. Regarding the dyslipidemic profile in BLOS1-hepatocyte-specific knockout mice, its clinical relevance may not be exactly mimicking patients with recessive *BLOC1S1* mutations as BLOS1 deficiency is an embryonic lethal (*Scott et al., 2014*; *Zhang et al., 2014*), and the whole body knockout of *BLOC1S1* could be different given some hypomorphic mutations are viable. Nevertheless, the involvement of BLOS1 in LDLR endosomal recycling may provide insights into the general regulatory mechanism of LDLR trafficking and the understanding of impaired LDL clearance in the liver.

## Materials and methods

### Key resources
All resources used in this study are listed in Appendix 1.

### Animals
The *Bloc1s1* floxed mice (loxp mice) and *Kxd1*-KO mice were originally generated in our lab (*Yang et al., 2012*; *Zhang et al., 2014*) and subsequently bred locally in the animal facility (specified pathogen free) of Institute of Genetics and Developmental Biology, Chinese Academy of Sciences. *pa* mice were derived from The Jackson Laboratory. *Alb*-Cre tool mice were supplied by Model Animal Research Center of Nanjing University. To generate liver-specific *Bloc1s1* knockout mice (cKO mice), *Alb*-Cre mice were crossed with loxp mice. Colonies were maintained by breeding *Alb*-Cre; loxp/loxp mice with loxp/loxp mice. Control loxp mice and cKO mice were littermates. Male mice of 3 months were used unless is stated. Mice were kept under a 12 hr dark-light period and provided a standard chow diet. For starvation treatment, mice were fasted for 12 hr overnight, and all mice livers used for Oil Red O staining were obtained at the next morning. All animal work were approved by the Institutional Animal Care and Use Committee of the Institute of Genetics and Developmental Biology, Chinese Academy of Sciences (mouse protocol KYD2005-006).

### Isolation and culture of mouse primary hepatocytes
A two-step collagenase perfusion technique was used to isolate primary hepatocytes from loxp and cKO mice liver as described (*Li et al., 2010*). Mice were euthanized by $CO_2$ inhalation, the abdominal cavity was opened, and then the precaval vein was closed with a vascular clamp. Liver perfusion from the portal vein was initiated with 50 mL pre-warmed (37°C) HBSS (pH 7.4, no calcium and magnesium) containing 10 mM HEPES and 200 μM EDTA for 10 min. After the liver turned pale, change perfusion medium and perfuse with 50 mL pre-warmed 50 HBSS (pH 7.4, with calcium and magnesium) containing 0.5 mg/mL collagenase IV and 20 mM HEPES for 10–15 min. Then remove the entire liver to a petri dish containing HBSS (with calcium and magnesium) on ice, and dissociate the liver lobs by tearing with forceps. The resulting cell suspension was filtered through a 70 μm mesh filter and washed three times with pre-cooled HBSS (centrifuged at $50 \times g$ for 2 min). Cells were then suspended in 10 mL William's E Medium (Gibco) supplemented with 10% (v/v) fetal bovine serum (FBS, Gibco), 1% (v/v) 200 mM GlutaMAX (Gibco) and 1% (v/v) Nonessential amino acids (NEAA, Gibco), and then seeded at a final density of $0.4 \times 10^6$ cells per mL onto collagen I-coated dishes (coverslips). Cells were incubated with 5% $CO_2$ at 37°C for 2 hr and then remove the medium

containing attached cells and replaced with fresh culture medium. Transfections were done 4 hr after attachment using jetPEI-Hepatocyte reagent (Polyplus).

## Culture of cell lines

Hep G2 cells were obtained from the cell bank of the Chinese Academy of Sciences (Shanghai, China). All cell lines used in experiments were maintained at 37°C with 5% $CO_2$. Hep G2 cells were cultured in Minimal Essential Medium (MEM, Hyclone) supplemented with 10% (v/v) FBS, 1% 100 mM sodium pyruvate (Gibco) and 1% NEAA (Gibco).

## GST-fusion protein purification

The strain used for protein expression and purification was *Escherichia coli* (*E. coli*) BL21 (DE3) (Vazyme), and the expressing vector was PGEX-4T-1 (Pharmacia). First, BL21 competent cells were transformed with the expression plasmid and selected on LB plates supplemented with 100 µg/mL ampicillin overnight at 37°C. Single colonies were picked and grown overnight in 5 mL LB medium supplemented with 100 µg/mL ampicillin at 37°C with shaking at 300 rpm. On next morning, the overnight culture was diluted 1:100 into fresh 100 mL LB medium and cultured at 37°C (about 2.5 hr) until the optical density at 600 nm ($OD_{600}$) reached 0.5 to 0.7. Isopropyl-1-thio-β-D-galactopyranoside (IPTG) was added to final concentration of 0.4 mM and fusion protein expression induced at 25°C for 8 hr.

## Constructs

For the expression of N-terminal tagged FLAG/Myc-fusion proteins (BLOS1-FLAG, BLOS1-Myc), pCMV-Tag-2B/3B vectors (Agilent) were used with cDNA fragments inserted into multiple cloning sites. pEGFP-C2 (Clontech) was used to generate BLOS1-GFP-C2 and RAB11A-GFP plasmids. pEGFP-N2 (Clontech) was used for KIF13A-GFP, KIF13A-ST-GFP expression. For the red fluorescent protein Scarlet (*Bindels et al., 2017*) expression, ORF of GFP was substituted by the coding sequence of Scarlet by modifying pEGFP-C2 and pEGFP-N2 vectors without changing other DNA elements. Similarly, GFP ORF in the pEGFP-N2 vector was replaced by FLAG/Myc/HA coding sequences to generate plasmid for C-terminal fusion proteins (BLOS1-HA, LDLR-HA, KIF13A-FLAG and all other KIF proteins with corresponding FLAG/Myc/HA tags) expression. The expressing plasmids of KIF protein rigor mutants were generated by site-directed mutagenesis using primers shown in Appendix 1.

The shRNA expressing plasmids used in stable knockdown cell line (*BLOC1S1*-KD, *KIF13A*-KD, and *KIF3A*-KD cells) construction were modified from pSilencer 5.1-H1 Retro vector (Ambion) by introducing a MfeI cleavage site (same overhang with EcoRI) between 2166 to 2172 just after the XhoI site and upstream of the expressing element of shRNA. Take KIF3A knockdown plasmid as an example, three separate shRNA expressing plasmids were generated by annealing the templates (Appendix 1) and inserting into the modified plasmids. One plasmid was digested with XhoI and EcoRI to get the whole shRNA expressing element, another was digested with XhoI and MfeI. Then ligation was performed to generate a new plasmid containing two sets of shRNA expressing elements. This procedure was repeated until all templates for a specific gene were introduced on one plasmid.

ORF of KIF5B was amplified from the cDNA of Hep G2 cells, and all the other ORFs were amplified from the cDNA obtained by reverse transcription of total mRNA of mouse liver. For the construction of KIF3A-FLAG-KIF3B-Myc-BLOS1-HA co-overexpression plasmid, the expression cassette of KIF3B-Myc was amplified and then introduced into BspTI digested KIF3A-FLAG plasmid by recombination (ClonExpress II One Step Cloning Kit, Vazyme) after which the BspTI site was kept intact, and the expression cassette of BLOS1-HA was inserted by a similar procedure.

## Immunoblotting

Homogenates of either cell cultures or tissues were used in western blot analysis. Cells in one well of 6 well plates were collected in ice-cold PBS (Gibco) and lysed in 200 µL lysis buffer (50 mM Tris-HCl pH 7.0, 150 mM NaCl, 1 mM EDTA, 1% Triton X-100) supplemented with protease inhibitor cocktail. Tissue samples (about 50 mg) were homogenized with a micro tissue grinder in 500 µL lysis buffer. All homogenates were rotated for 1 hr at 4°C in a vertical rotator and centrifuged at 13000 × *g* for

15 min. Supernatants with soluble proteins were mixed with 6 × Laemmli loading buffer with β-mercaptoethanol and boiled for 5 min. Protein samples were resolved by SDS-PAGE on 10% or 15% Tris-Glycine buffered polyacrylamide gels and transferred to PVDF membranes using mini trans-blot module (Bio-Rad).

The blots were blocked in 5% (w/v) non-fat milk (BD) at room-temperature for 1 hr in PBS/0.1% (v/v) Tween 20 (PBST). After a brief wash in PBST, membranes were then incubated in primary antibody diluted in 3% (w/v) BSA dissolved in PBST overnight at 4°C. On the following day, the membranes was washed three times, each time for 10 min in PBST with shaking and then incubated with 1:5000 HRP-conjugated secondary antibodies (Zsbio) diluted in blocking buffer at room temperature for 1 hr. Three more 10 min washes with PBST were then performed before detection using Chemiluminescent Substrate (Thermo) and imaging with a Minchemi system (Sage Creation). Quantification was performed using Fiji (*Schindelin et al., 2012*). The following primary antibodies were used (additional information was shown in the Appendix 1): anti-LDLR (1/5000 for ab52818; 1/2000 for ab30532), anti-TfR (1/2000), anti-β-Actin (1/50000), anti-KIF13A (1/1000), anti-KIF3A (1/1000), anti-Pallidin (1/2000), anti-Dysbindin (1/20000), anti-GST (1/5000), anti-FLAG (Sigma, Cat#F3165, 1/5000), anti-Myc (MBL, Cat#562, 1/2000), anti-HA (Abcam, 1/5000).

## Co-immunoprecipitation

All immunoprecipitations were performed in the absence of cross-linking reagents. Cells in six well plates were transfected with FLAG empty vector/FLAG fusion protein plasmid and corresponding candidates of interacting protein plasmids for 48 to 72 hr, and cell lysates were prepared in cell lysis buffer as described above. 40 μL anti-FLAG M2 agarose beads (Sigma) were used for the enrichment of FLAG-fusion proteins of each well by incubating with the supernatants overnight at 4°C on a vertical rotator. Sufficient washes were carried out at 4°C using lysis buffer with a total time of 30 min on a vertical rotator (5 to 6 washes), beads were pelleted by centrifuging at 5000 × *g* for 30 s after each wash. Proteins were eluted by boiling with 50 μL 2 × Laemmli loading buffer for 5 min and subsequent western blot was performed as described above. For the detection of eluted FLAG-fusion proteins, a particular HRP-conjugated secondary antibody (Abcam, Cat# ab131368) which preferentially detects the non-reduced form of mouse IgG over the reduced, SDS-denatured forms was used to eliminate the IgG band results from the anti-FLAG M2 beads.

## GST pull-down assay

The full length and truncated BLOS1s were expressed in *E. coli* BL21 as described above, and GST protein which was induced at 37°C for 4 hr served as control. After induction, bacteria were collected by centrifuge at 4000 × *g*, 4°C for 15 min. The bacteria pellet was resuspended in 10 mL ice-cold lysis buffer (50 mM Tris-HCl pH 8.0, 150 mM NaCl, 5 mM EDTA, 1% Triton X-100) supplemented with protease inhibitor and 1 mM DTT. The suspension was then sonicated on ice using a probe-tip sonicator using the following parameters: 10 s each with 30 s rest at 200 W for a total time of 15 min. The supernatant was collected and stored at −20°C after centrifugation at 12,000 × *g*, 4°C for 15 min.

For GST pull-down assay, 50 μL GST protein-containing supernatant diluted in 450 μL lysis buffer was used to incubate with 50 μL slurry of glutathione-Sepharose beads at 4°C for 6 hr, and the dosage of supernatants of other GST fusion proteins was determined by Coomassie brilliant blue staining. The beads were pelleted at 5000 × *g* for 30 s and washed five to six times with lysis buffer before the incubation with 500 μL mouse liver lysate overnight at 4°C. After incubation, beads were washed with tissue lysis buffer for another 5 to 6 times, and binding proteins were eluted with 2 × Laemmli loading buffer by boiling for 5 min. Immunoblots were performed as described above.

## Oil Red O staining of mouse liver sections

Freshly collected liver lobes of control (loxp) and cKO mice fed in chow diets or under starvation were embedded in OCT (Tissue-Tek) and flash-frozen immediately with liquid nitrogen. Samples were left in the chamber of the freezing microtome with the temperature set at −20°C for at least 1 hr before the sectioning. Three consecutive 10-μm-thick sections of the liver were collected onto one slide and fixed immediately in ice-cold 4% (w/v) paraformaldehyde dissolved in PBS (pH 7.4) for 10 min and then rinsed in three changes of distilled water. Slides were dehydrated in 60% isopropyl

alcohol for 2 min before the immersed by Oil Red O staining solution (1.5 part of Oil Red O saturated isopropyl alcohol solution mixed with one part of distilled water) at room temperature for 15 min. Slides were rinsed in 60% isopropyl alcohol for two times and then placed in distilled water before the next step. Counterstaining of the nucleus with Mayer's hematoxylin was done by submerging the sections in hematoxylin for 10 s and thereafter dipping the sections in distilled water for three times before bluing the stain in PBS (pH 7.2) for 5 min. Slides were mounted in water-soluble mounting medium (PBS: glycerol = 1:9), and coverslip edges were sealed with nail polish. Micrographs were acquired using an Olympus DP71 imaging system, and Fiji was used in the quantification of Oil Red O staining areas.

## Immunocytochemistry and immunofluorescence imaging

Coverslips were pre-coated with collagen I (Sigma) for Hep G2 attachment. Cells attached on coverslips were fixed in 4% (w/v) formaldehyde dissolved in PBS and immunocytochemistry (ICC) was performed according to a general ICC protocol (Abcam). Permeabilization of cells was done by incubating in PBS containing 0.1% (v/v) Triton X-100 for 15 min. Blocking was done with 3% (w/v) BSA in PBS. The following primary antibodies were used (additional information was shown in Appendix 1): anti-mouse LDLR (R and D, Cat#AF2255, 1/100), anti-human LDLR (R and D, Cat#AF2148, 1/100), anti-TfR (1/200), anti-EEA1 (1/500), anti-Cytochrome C (1/250), anti-CD63 (1/200), anti-FLAG (Sigma, Cat#F3165, 1/1000; Sigma, Cat#F7425, 1/1000), anti-Myc (MBL, Cat#562, 1/500; MBL, Cat#M192-3S, 1/500), anti-HA (Santa Cruz, Cat#sc-7392, 1/100; Abcam, Cat#ab9134, 1/1000) and anti-α-Tubulin (Abcam, Cat#ab7291, 1/500; Abcam, Cat#ab18251, 1/1000). Coverslips were incubated with primary antibodies diluted in blocking buffer overnight at 4°C and wash buffer used after antibody incubation was PBS containing 0.1% (v/v) Tween 20. All Alexa Fluor-conjugated secondary antibodies (Invitrogen/Abcam) were diluted 1/1000 and used with an incubating time of 1 hr at 37°C in the dark. After the wash, coverslips were mounted on glass slides with Prolong Gold Antifade Mountant (Invitrogen), sealed with nail polish and stored in the dark at 4°C before imaging. Confocal imaging was carried out using a Nikon Eclipse Ti Confocal Laser Microscope System with NIS-Elements Software (Nikon, Japan). Super-resolution images were acquired in a Zeiss LSM 880 system with Airyscan module. Images were analyzed and quantified were Fiji software with raw data imported through Bio-Formats Importer.

## Electrophoretic separation of plasma lipoproteins in native gradient polyacrylamide gel

Lipoproteins in plasma samples were pre-stained with Sudan Black B (SBB) staining solution (SBB saturated thanol solution) by mixing 2 μL of SBB staining solution with 30 μL plasma and incubating at 37°C for 30 min. A 4% to 15% polyacrylamide gradient mini gel prepared with Tris-HCl buffer (pH 8.3) was used to separate lipoproteins, and the running buffer (0.6 g/L Tris, 2.88 g/L Glycine, pH 8.3) was changed several times during the overnight electrophoresis at 4°C with a voltage of 70 V. For the Oil Red O staining, unstained plasma samples were separated by electrophoresis and then stained with 0.2% (w/v) Oil Red O methanol solution for 30 min with shaking.

## Isolation of plasma LDL and LDL-DiI endocytosis assay

LDL of pooled mouse plasma were isolated by sequential ultracentrifugation as described (*Havel et al., 1955*). High-density salt solution (1.346 g/mL) was prepared by dissolving 153 g NaCl and 354 g KBr in Milli-Q water to a total volume of 1L. Low-density salt solution (1.005 g/mL) contained 0.15 M NaCl. Salt solutions of other densities were prepared by mixing the high-density and low-density salt solutions at different ratios. 5 mL pooled plasma with the density of 1.006 g/mL was mixed with 1 mL of 1.085 g/mL salt solution to reach a total density of 1.019 g/mL and ultracentrifuged at $180,000 \times g$, 4°C for 12 hr. The top 1.5 mL layer containing lipoproteins with a density less than 1.019 g/mL (mainly VLDL) was collected, and additional 1.5 mL salt solution (1.200 g/mL) was mixed with the remaining plasma to reach a density of 1.063 g/mL. Ultracentrifugation was performed as above, and the top 1.5 mL solution containing lipoproteins with the density between 1.019 g/mL and 1.063 g/mL (LDL) was collected. All collected lipoproteins were dialyzed in 500 mL 0.15 M NaCl twice at 4°C before electrophoresis or DiI labeling.

For DiI labeling, 1 mg of purified LDL were mixed with 50 μL DiI stock solution (3 mg/mL in DMSO; Sigma) and incubated at 37°C for 8 hr. The density was adjusted to 1.063 g/mL, and ultra-centrifugation was performed. Collected LDL-DiI on the top layer was dialyzed with 0.15 M NaCl and further filter-sterilized (0.1 μm, Millipore). LDL-DiI were added to the medium of primary hepatocytes at the final concentration of 10 μg/mL, and binding of LDL-DiI to the cell surface was done at 4°C for 30 min. After the binding, cells at different time points post endocytosis were fixed and visualized.

### Generation of stable knockdown cell lines
Hep G2 cells in 24 well plates were transfected with targeted gene shRNA or scrambled negative control shRNA expressing plasmids and split into 60 mm dishes at the concentration of 10% confluent 24 hr post-transfection. The selection was started the day after splitting using 2 μg/mL puromycin (concentration determined by preliminary experiment). Culture medium was changed every day in the first week and every 3 days after that. Cells were grown for 3 weeks under selection pressure, and single colonies were picked and expanded in 24-well plates with puromycin added. Western blot of target proteins was performed to identify positive colonies.

### Inhibition of lysosomal degradation
Hep G2 cells were plated in 12-well plates at a seeding density of ~50% confluency and were cultured for 12 hr before leupeptin treatment. Leupeptin (Sigma) dissolved in Milli-Q water was added to a final concentration of 50 μM in the culture medium. Cells were collected 24 hr later, and western blot was performed with indicated antibodies as previously described.

### Live-cell imaging and super-resolution microscopy
Hep G2 cells were cultured on collagen I-coated 35 mm μ-Dish (*ibid*) with a high-performance glass bottom (170 ± 5 μm). Hep G2 cells were transfected with GFP or/and Scarlet fusion protein expressing plasmids using lipofectamine 3000 reagent 24 hr before the imaging. Live cell imaging was carried out on a Nikon Eclipse Ti Confocal Laser Microscope System equipped with 405 nm (20 mW), 488 (50 mW) and 561 (100 mW) laser lines, temperature controller, a 100 × oil immersion objective (Nikon, NA = 1.40) and appropriate filter sets. Before imaging, the culture medium was replaced by $CO_2$-independent Live Cell Imaging Solution (Invitrogen) and adapted for 15 min. Image series were captured using the 488 laser line (~10% laser power) and 561 laser line (~5% laser power) with 1 s interval and total imaging time of 120 s under the 'live' mode.

Super-resolution microscopy of microtubules was carried out on Zeiss LSM 880 system equipped with Airyscan module and cell incubation chamber. Cells transfected with RAB11A-GFP were first stained with Tubulin Tracker Deep Red according to the manufacturer's instruction, and image series were then captured using the 488 laser line and 633 laser line using Airyscan module with a interval time of 0.32 s and zoom factor of 3. Subsequent imaging processing was accomplished using ZEN 2.3 (Zeiss) and Fiji software.

### Quantification and statistical analysis
Data were presented as Mean ± SEM unless stated, the statistical significance of mean differences was determined using two-tailed Student's t test as indicated in the figure legends. The sample size (n) was also indicated in the corresponding figure legends, which represents the number of identically-treated replicates. Statistical significance is defined as, n.s., not significant, *$p < 0.05$, **$p < 0.01$, ***$p < 0.001$. Statistical analyses were performed using GraphPad Prism.

## Acknowledgements

We thank Prof. Richard T Swank for his proofreading this manuscript and comments. This work was partially supported by grants from the Ministry of Science and Technology of China (2019YFA0802104), and the National Natural Science Foundation of China (31830054; 91954104; 91539204; 81670789), and the Chinese Academy of Sciences (XDA12030211).

## Additional information

### Funding

| Funder | Grant reference number | Author |
| --- | --- | --- |
| Ministry of Science and Technology of the People's Republic of China | 2019YFA0802104 | Wei Li |
| National Natural Science Foundation of China | 31830054 | Wei Li |
| National Natural Science Foundation of China | 91954104 | Chanjuan Hao |
| National Natural Science Foundation of China | 91539204 | Wei Li |
| Chinese Academy of Sciences | XDA12030211 | Guanghou Shui |
| National Natural Science Foundation of China | 81670789 | Chanjuan Hao |

The funders had no role in study design, data collection and interpretation, or the decision to submit the work for publication.

### Author contributions

Chang Zhang, Conceptualization, Formal analysis, Investigation, Methodology, Writing - original draft; Chanjuan Hao, Data curation, Formal analysis, Funding acquisition; Guanghou Shui, Conceptualization, Data curation, Supervision, Funding acquisition, Methodology; Wei Li, Conceptualization, Data curation, Supervision, Funding acquisition, Validation, Writing - original draft, Project administration, Writing - review and editing

### Author ORCIDs

Chang Zhang (iD) https://orcid.org/0000-0002-3930-9799
Wei Li (iD) https://orcid.org/0000-0002-0248-5510

### Ethics

Animal experimentation: All animal work was approved by the Institutional Animal Care and Use Committee of the Institute of Genetics and Developmental Biology, Chinese Academy of Sciences (mouse protocol KYD2005-006).

### Decision letter and Author response

Decision letter https://doi.org/10.7554/eLife.58069.sa1
Author response https://doi.org/10.7554/eLife.58069.sa2

## Additional files

### Supplementary files

• Transparent reporting form

### Data availability

All data generated or analysed during this study are included in the manuscript and supporting files.

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

# Appendix 1

**Appendix 1—key resources table**

| Reagent type (species) or resource | Designation | Source or reference | Identifiers | Additional information |
|---|---|---|---|---|
| Genetic reagent (*M. musculus*) | *Kxd1*-KO | Prepared in our lab (*Yang et al., 2012*) | | |
| Genetic reagent (*M. musculus*) | *pa* | The Jackson Laboratory | JAX: 000024; RRID:IMSR_JAX: 000024 | |
| Genetic reagent (*M. musculus*) | *Alb-Cre* | Model animal research center of Nanjing university | RRID:IMSR_JAX: 003574 | Derived from The Jackson Laboratory |
| Genetic reagent (*M. musculus*) | *loxp* | Prepared in our lab (*Zhang et al., 2014*) | | |
| Genetic reagent (*M. musculus*) | *Bloc1s1*-cKO | This paper | | |
| Strain, strain background (*Escherichia coli*) | BL21(DE3) | Vazyme | C504 | Chemical competent cells |
| Strain, strain background (*Escherichia coli*) | DH5α | Vazyme | C502 | Chemical competent cells |
| Cell line (*Homo-sapiens*) | Hep G2 | Cell bank of Chinese Academy of Sciences (Shanghai, China) | Cat#TCHu72; RRID:CVCL_0027 | Has been authenticate by STR profiling and tested negative for mycoplasma in cell bank |
| Cell line (*Homo-sapiens*) | HEK293T | Cell bank of Chinese Academy of Sciences (Shanghai, China) | Cat#GNHu17; RRID:CVCL_0063 | Has been authenticate by STR profiling and tested negative for mycoplasma in cell bank |
| Biological sample (*M. musculus*) | Primary mouse hepatocytes | This paper | | Freshly isolated from mouse liver |
| Antibody | Mouse monoclonal anti-alpha Tubulin (clone DM1A) | Abcam | Cat#ab7291; RRID:AB_2241126 | IF (1:500) |
| Antibody | Rabbit polyclonal anti-alpha Tubulin | Abcam | Cat#ab18251; RRID:AB_2210057 | IF (1:1000) |
| Antibody | Rabbit polyclonal anti-Transferrin Receptor | Abcam | Cat#ab84036; RRID:AB_10673794 | IF (1:200), WB (1:2000) |
| Antibody | Rabbit polyclonal anti-LDL Receptor | Abcam | Cat#ab30532; RRID:AB_881272 | WB (1:2000) |
| Antibody | Rabbit monoclonal anti-LDL Receptor (clone EP1553Y) | Abcam | Cat#ab52818; RRID:AB_881213 | WB (1:5000) |
| Antibody | Rabbit polyclonal anti-PCSK9 | Abcam | Cat#ab31762; RRID:AB_777140 | WB (1:1000) |

*Continued on next page*

*Appendix 1—key resources table continued*

| Reagent type (species) or resource | Designation | Source or reference | Identifiers | Additional information |
|---|---|---|---|---|
| Antibody | Goat polyclonal anti-HA tag antibody | Abcam | Cat#ab9134; RRID:AB_307035 | IF (1:1000), WB (1:5000) |
| Antibody | Rat monoclonal anti-mouse IgG for IP (HRP) | Abcam | Cat#ab131368; N/A | WB (1:5000) |
| Antibody | Donkey anti-rabbit Alexa Fluor 405 | Abcam | Cat#ab175649; AB_2715515 | IF (1:1000) |
| Antibody | Mouse monoclonal anti-EEA1 (clone 14) | BD | Cat#610457; RRID:AB_397830 | IF (1:500) |
| Antibody | Mouse monoclonal anti-Cytochrome C (clone 6H2.B4) | BD | Cat#556432; RRID:AB_396416 | IF (1:250) |
| Antibody | Mouse monoclonal anti-CD63 (clone H5C6) | BD | Cat#556019; RRID:AB_396297 | IF (1:200) |
| Antibody | Goat polyclonal anti-mouse LDL Receptor | R and D | Cat#AF2255; RRID:AB_355203 | IF (1:100) |
| Antibody | Goat polyclonal anti-human LDL Receptor | R and D | Cat#AF2148; RRID:AB_2135126 | IF (1:100) |
| Antibody | Mouse monoclonal anti-beta Actin (clone AC-15) | Sigma-Aldrich | Cat#A5441; RRID:AB_476744 | WB (1:50000) |
| Antibody | Mouse monoclonal ant-Acetylated Tubulin antibody (clone 6-11B-1) | Sigma-Aldrich | Cat#T7451; RRID: AB_609894 | IF (1:200) |
| antibody | Mouse monoclonal anti-FLAG tag antibody (clone M2) | Sigma-Aldrich | Cat#F3165; RRID: AB_259529 | IF (1:1000), WB (1:5000) |
| Antibody | Rabbit polyclonal anti-FLAG tag antibody | Sigma-Aldrich | Cat#F7425; RRID:AB_439687 | IF (1:1000) |
| Antibody | Rat monoclonal anti-Tyrosinated Tubulin antibody (clone YL1/2) | Millipore | Cat#MAB1864; RRID:AB_2210391 | IF (1:200) |
| Antibody | Rabbit monoclonal anti-KIF3A antibody (clone D7G3) | Cell Signaling Technology | Cat#8507; RRID:AB_11141049 | WB (1:1000) |
| Antibody | Rabbit polyclonal anti-KIF13A antibody | Bethyl Laboratories | Cat#A301-077A; RRID:AB_873053 | WB (1:1000) |
| Antibody | Rabbit polyclonal anti-Myc tag antibody | MBL | Cat#562; RRID:AB_591105 | IF (1:500) |
| Antibody | Mouse monoclonal anit-Myc tag antibody (clone Myc3) | MBL | Cat#M192-3S; RRID:AB_11161202 | IF (1:500), WB (1:2000) |
| Antibody | Mouse monoclonal anti-HA tag antibody (clone F-7) | Santa Cruz | Cat#sc-7392; RRID:AB_627809 | IF (1:100) |
| Antibody | Mouse monoclonal anti-GST antibody (clone B-14) | Santa Cruz | Cat#sc-138; RRID:AB_627677 | WB (1:5000) |
| Antibody | Rabbit polyclonal anti-Pallidin antibody | Proteintech | Cat#10891–2-AP; RRID:AB_2164174 | WB (1:2000) |

*Continued on next page*

*Appendix 1—key resources table continued*

| Reagent type (species) or resource | Designation | Source or reference | Identifiers | Additional information |
|---|---|---|---|---|
| Antibody | Rabbit polyclonal anti-Dysbindin antibody | Prepared in our lab *Wang et al., 2014* | N/A | WB (1:20000) |
| Antibody | Donkey anti-mouse Alexa Fluor 488 | ThermoFisher | Cat#A-21202; RRID:AB_141607 | IF (1:1000) |
| Antibody | Donkey anti-mouse Alexa Fluor 594 | ThermoFisher | Cat#A-21203; RRID:AB_2535789 | IF (1:1000) |
| Antibody | Donkey anti-Rabbit Alexa Fluor 488 | ThermoFisher | Cat#A-21206; RRID:AB_2535792 | IF (1:1000) |
| Antibody | Donkey anti-Rabbit Alexa Fluor 594 | ThermoFisher | Cat#A-21207; RRID:AB_141637 | IF (1:1000) |
| Antibody | Donkey anti-Goat Alexa Fluor 488 | ThermoFisher | Cat#A-11055; RRID:AB_2534102 | IF (1:1000) |
| Antibody | Donkey anti-Goat Alexa Fluor 594 | ThermoFisher | Cat#A-11058; RRID:AB_2534105 | IF (1:1000) |
| Recombinant DNA reagent | BLOS1-GFP-C2 (plasmid) | This paper | | |
| Recombinant DNA reagent | BLOS1-GFP-N2 | This paper | | |
| Recombinant DNA reagent | BLOS1-FLAG | This paper | | |
| Recombinant DNA reagent | BLOS1-Myc | This paper | | |
| Recombinant DNA reagent | BLOS1-HA | This paper | | |
| Recombinant DNA reagent | GST-BLOS1 | This paper | | |
| Recombinant DNA reagent | RAB11A-GFP-C2 | This paper | | |
| Recombinant DNA reagent | RAB11A-Scarlet-C2 | This paper | | |
| Recombinant DNA reagent | KIF13A-FLAG | This paper | | |
| Recombinant DNA reagent | KIF13A-GFP-N2 | This paper | | |
| Recombinant DNA reagent | KIF13A-ST-GFP-N2 | This paper | | |
| Recombinant DNA reagent | KIF13A-HA | This paper | | |
| Recombinant DNA reagent | KIF13A-R-HA | This paper | | |
| Recombinant DNA reagent | KIF13A-R-Scarlet-N2 | This paper | | |
| Recombinant DNA reagent | KIF3B-Myc | This paper | | |
| Recombinant DNA reagent | KIF3B-R-Myc | This paper | | |

*Continued on next page*

*Appendix 1—key resources table continued*

| Reagent type (species) or resource | Designation | Source or reference | Identifiers | Additional information |
|---|---|---|---|---|
| Recombinant DNA reagent | KIF3B-R-Scarlet-N2 | This paper | | |
| Recombinant DNA reagent | KIF3A-FLAG | This paper | | |
| Recombinant DNA reagent | KIF3A-R-FLAG | This paper | | |
| Recombinant DNA reagent | KIF3C-HA | This paper | | |
| Recombinant DNA reagent | KIF3C-R-HA | This paper | | |
| Recombinant DNA reagent | KAP3-HA | This paper | | |
| Recombinant DNA reagent | KIF5B-R-Myc | This paper | | |
| Recombinant DNA reagent | KIF16B-FLAG | This paper | | |
| Recombinant DNA reagent | KIF3A-FLAG-KIF3B-Myc-BLOS1-HA | This paper | | |
| Sequence-based reagent | LDLR_1F | This paper | RT-PCR primers | GTCTTGGCACTGGAACTCGT |
| Sequence-based reagent | LDLR_1R | This paper | RT-PCR primers | CTGGAAATTGCGCTGGAC |
| Sequence-based reagent | LDLR-2F | This paper | RT-PCR primers | ACGGCGTCTCTTCCTATGACA |
| Sequence-based reagent | LDLR-2R | This paper | RT-PCR primers | CCCTTGGTATCCGCAACAGA |
| Sequence-based reagent | GAPDH-F | This paper | RT-PCR primers | GGAGCGAGATCCCTCCAAAAT |
| Sequence-based reagent | GAPDH-R | This paper | RT-PCR primers | GGCTGTTGTCATACTTCTCATGG |
| Sequence-based reagent | KIF13A-R-F | This paper | Site-directed mutagenesis primers | GAGCCTGGTAGACCTGGCGGCGAGCGAGAGAGTGTCGAAGAC |
| Sequence-based reagent | KIF13A-R-R | This paper | Site-directed mutagenesis primers | GTCTTCGACACTCTCTCGCTCGCCGCCAGGTCTACCAGGCTC |
| Sequence-based reagent | KIF3B-R-F | This paper | Site-directed mutagenesis primers | CTGAATCTTGTAGATCTTGCTGCCAGTGAGCGGCAAGCCAAG |
| Sequence-based reagent | KIF3B-R-R | This paper | Site-directed mutagenesis primers | CTTGGCTTGCCGCTCACTGGCAGCAAGATCTACAAGATTCAG |

*Continued on next page*

*Appendix 1—key resources table continued*

| Reagent type (species) or resource | Designation | Source or reference | Identifiers | Additional information |
|---|---|---|---|---|
| Sequence-based reagent | KIF3C-R-F | This paper | Site-directed mutagenesis primers | GTAGACCTGGCCGCC AGTGAGAGACAG |
| Sequence-based reagent | KIF3C-R-R | This paper | Site-directed mutagenesis primers | CTGTCTCTCACTGGC GGCCAGGTCTAC |
| Sequence-based reagent | KIF5B-R-F | This paper | Site-directed mutagenesis primers | CTGGTTGATTTAGCTGC TAGTGAAAAGGTTAG |
| Sequence-based reagent | KIF5B-R-R | This paper | Site-directed mutagenesis primers | CTAACCTTTTCACTAGCA GCTAAATCAACCAG |
| Sequence-based reagent | MfeI-F | This paper | Site-directed mutagenesis primers for pSilencer 5.1-H1 Retro vector | ATGGAGGACCCCAA TGCCAAGG |
| Sequence-based reagent | MfeI-R | This paper | Site-directed mutagenesis primers for pSilencer 5.1-H1 Retro vector | CCGAGTGGCTGTGGC TTCC |
| Sequence-based reagent | BLOS1-1F | This paper | shRNA template primers | gatccgTCGGAATGG TGGAGA ACTTgagaAAGTTC TCCACCA TTCCGAtttttttggaaa |
| Sequence-based reagent | BLOS1-1R | This paper | shRNA template primers | agcttttccaaaaaaTCGGAA TG GTGGAGAACTTtctcAAG TTC TCCACCATTCCGAcg |
| Sequence-based reagent | BLOS1-2F | This paper | shRNA template primers | gatccGCACTGGAATATG TCTACAgaga TGTAGACATATTCCAG TGCttttttggaaa |
| Sequence-based reagent | BLOS1-2R | This paper | shRNA template primers | agcttttccaaaaaaGCAC TGGAATATGTC TACAtctcTGTAGACATA TTCCAGTGCg |
| Sequence-based reagent | BLOS1-3F | This paper | shRNA template primers | gatccgCAGAAGCTTTGG TGGATCAgaga TGATCCACCAAAGCTTC TGtttttttggaaa |
| Sequence-based reagent | BLOS1-3R | This paper | shRNA template primers | agcttttccaaaaaaCAGAAGC TTTGGTG GATCAtctcTGA TCCACCAAAGCTTCTGcg |
| Sequence-based reagent | KIF13A-1F | This paper | shRNA template primers | cggGGAAACC TCCCAAGGTATTTGgaga CAAATACCTTGGGAGG TTTCCttttttga |

*Continued on next page*

*Appendix 1—key resources table continued*

| Reagent type (species) or resource | Designation | Source or reference | Identifiers | Additional information |
|---|---|---|---|---|
| Sequence-based reagent | KIF13A-1R | This paper | shRNA template primers | agcttCAAAAAGGAAACC TCCCAAGGTAT TTGtctcCAAATACC TTGGGAGGTTTCC |
| Sequence-based reagent | KIF13A-2F | This paper | shRNA template primers | ccggTTAACGAACTTC TGGTTTATTgaga AATAAACCAGAAGTTCG TTAAtttttga |
| Sequence-based reagent | KIF13A-2R | This paper | shRNA template primers | agcttCAAAAA TTAACGAACTTCTGGTT TATTtctcAA TAAACCAGAAGTTCG TTAA |
| Sequence-based reagent | KIF3A-1F | This paper | shRNA template primers | ccggCGTCAGTCTTTGA TGAAACTAgaga TAGTTTCATCAAAGAC TGACGtttttga |
| Sequence-based reagent | KIF3A-1R | This paper | shRNA template primers | agcttCAAAAACGTCAGTC TTTGATGAA ACTAtctcTAGTTTCA TCAAAGACTGACG |
| Sequence-based reagent | KIF3A-2F | This paper | shRNA template primers | ccggGCCTG TTTGAACACATTCTAA gaga TTAGAATGTG TTCAAACAGGCtttttga |
| Sequence-based reagent | KIF3A-2R | This paper | shRNA template primers | agcttCAAAAAGCCTG TTTGAACACATTC TAAtctcTTAGAATGTG TTCAAACAGGC |
| Sequence-based reagent | KIF3A-3F | This paper | shRNA template primers | ccggCGGGATTA TCAGGAAATGATTga gaAATCATTTCCTGATAA TCCCGtttttga |
| Sequence-based reagent | KIF3A-3R | This paper | shRNA template primers | agcttCAAAAACGGGATTA TCAGGAAAT GATTtctcAATCATTTCC TGATAATCCCG |
| Peptide, recombinant protein | Streptavidin | Thermo Fisher | Cat. #: 434302 | |
| Commercial assay or kit | iScript cDNA Synthesis Kit | BIO-RAD | 1708891 | |
| Commercial assay or kit | RNeasy Mini Kit | QIAGEN | 74104 | |
| Commercial assay or kit | ClonExpress II One Step Cloning Kit | Vazyme | C112 | |
| Chemical compound, drug | Puromycin | InvivoGen | ant-pr-1 | |
| Chemical compound, drug | Leupeptin | Sigma-Aldrich | L5793 | |

*Continued on next page*

*Appendix 1—key resources table continued*

| Reagent type (species) or resource | Designation | Source or reference | Identifiers | Additional information |
|---|---|---|---|---|
| Chemical compound, drug | Oil Red O | Sigma-Aldrich | O9755 | |
| Chemical compound, drug | Sudan Black B | Sigma-Aldrich | 199664 | |
| Chemical compound, drug | 1,1'-Dioctadecyl-3,3,3',3'-tetramethylindocarbocyanine perchlorate (DiI) | Sigma-Aldrich | 42364 | |
| Software, algorithm | Fiji | http://fiji.sc/; *Schindelin et al., 2012* | RRID:SCR_002285 | Version 2.0.0-rc-69/1.52 n |
| Other | Minimal Essential Medium (MEM) | GE Healthcare | SH30024.01 | |
| Other | Glutathione Sepharose 4B resin | GE Healthcare | 17075601 | |
| Other | Collagen, Type I | Sigma-Aldrich | C3867 | |
| Other | Collagenase, TypeIV | Sigma-Aldrich | C5138 | |
| Other | Anti-FLAG M2 affinity gel | Sigma-Aldrich | A2220 | |
| Other | Phalloidin Alexa Fluor 594 | ThermoFisher | A12381 | |
| Other | Prolong Gold Antifade Mountant | ThermoFisher | P36935 | |
| Other | Lipofectamine 3000 | ThermoFisher | L3000015 | |
| Other | Sodium pyruvate | ThermoFisher | 11360070 | |
| Other | MEM Non-Essential Amino Acids | ThermoFisher | 11140050 | |
| Other | Tubulin Tracker Deep Red | ThermoFisher | T34076 | |
| Other | jetPEI-Hepatocyte | Polyplus | 102–05N | |

Note: The listed references in this table can be referred to the reference list in main text.

