## [Decision Letter]

**Acceptance summary:**

This paper provides convincing evidence that BLOS1 is a key regulator of endosomal recycling of the LDL receptor. Depletion of BLOS1 in mouse liver leads to reduced hepatocyte lipid droplets and elevated plasma LDL level, thus revealing an unexpected link between endosomal trafficking and lipid metabolism. Moreover, this study showed that BLOS1 acts as an adaptor for kinesin-2, which could be a new function of BLOS1 independent of BLOC1 and BORC complexes.

**Decision letter after peer review:**

Thank you for submitting your article "BLOS1 mediates kinesin switch during endosomal recycling of LDL receptor" for consideration by *eLife*. Your article has been reviewed by three peer reviewers, one of whom is a member of our Board of Reviewing Editors, and the evaluation has been overseen by a Reviewing Editor and Anna Akhmanova as the Senior Editor.

The reviewers have discussed the reviews with one another and the Reviewing Editor has drafted this decision to help you prepare a revised submission.

Summary:

In this manuscript, the authors described a defect of LDL clearance in liver-BLOS1-KO mice and found it was caused by reduced LDLR on cell surface due to the interrupted recycling endosome transport mediated by motor kinesin-3 and kinesin-2. Authors propose that BLOS1 served as the adaptor to switch the motors at microtubule intersections for recycling endosome long-range anterograde transport to cell periphery.

Essential revisions:

1) The distribution of LDLR and KIF3A should be shown in BLOS1 KO cells.

2) The co-localization of BLOS1 with LDLR, Rab 11A, and KIFs should be examined in primary hepatocytes.

3) Figure 4A: The authors showed the peripheral distribution of LDLR in the WT primary hepatocytes. Since BLOS1 KO primary hepatocytes showed reduced uptake of LDL via LDLR (Figure 2A,C), the authors should examine the distribution of LDLR in BLOS1 KO primary hepatocytes. The authors should also perform a series of transfection studies on WT and cKO primary hepatocytes (e.g. BLOS1 [rescue study on cKO], KIF13A [requirement of BLOS1 for KIF13A to bind to RE], KIF3A [requirement of BLOS1 for KIF3A to bind to LDLR] and so on).

4) Figure 4H/Figure 5C: In KIF13A-R (Figure 4H) or KIF3B-R (Figure 5C) over-expressed cells, KIF13A-R or KIF3B-R fully decorated the surface of non-physiologically bundled MTs. Therefore, the authors should exclude the possibility that microtubule-based motor protein cannot use such MTs as traffic rails. Authors should use another rigor mutant motor as a negative control, such as KIF5B-R or KIF16B-R, which does not interact with BLOS1.

Reviewer #1:

In this manuscript, authors reported BLOS1 is required for endosomal recycling of LDLR. Authors further shown BLOS1 act as adaptor protein for kinesin-2, moreover, they shown BLOS1 coordinate kinesin-3 and kinesin-2 for long-range transport of recycling endosome to pass microtubules intersections. This is an interesting study and overall, data are compelling. I have a few suggestions to further strength this study.

1) The majority of conclusion are draw from imaging analysis on cells overexpressing various plasmids, although is not absolutely required, biochemical analysis using endogenous antibody can significantly strength their case, for example, author can purified the recycling endosome from control and BLOS1 KO mouse and compare the sorting of LDLR and the recruitment of kinesin-3 and kinesin-2 to recycling endosome.

2) Authors propose BLOS1 is an adaptor for kinesin-2, author need provide direct evidence by shown the recruitment of kinesin-2 to recycling endosome is reduced in BLOS1 KD or KO cells.

3) The interaction between BLOS1 and kinesin-2 and kinesin-3 need to be confirmed by endogenous IP.

Reviewer #2:

In this manuscript, the authors described a defect of LDL clearance in liver-BLOS1-KO mice and found it was caused by reduced LDLR on cell surface due to the interrupted recycling endosome transport mediated by motor kinesin-3 and kinesin-2. BLOS1 served as the adaptor to switch the motors at microtubule intersections for recycling endosome long-range anterograde transport to cell periphery. The novelty of this manuscript is BLOS1-caused dyslipidemia and BLOS1 acts as an adaptor for kinesin-2, which could be new functions of BLOS1 independent of BLOC1 and BORC complexes. The manuscript was well written, and most figures were well organized to convey the information. To make the hypothesis more convincing, there are several concerns:

1) The authors have shown PCSK9 level in the liver tissue of cKO mice (Figure 2—figure supplement 1B). PCSK9 is a secreted protein and initiates the function to LDLR in plasma. Could the authors also test PCSK9 level in the plasma?

2) The authors have nicely shown LDLR is tightly associated with KIF13A (Figure 4C and E). Overexpressing KIF13A could accumulate the cargos LDLR and TfR at cell tips (Figure 3B, Figure 4B and D). It looks like KIF13A did not closely associate with BLOS1 (Figure 5D), and overexpressing KIF13A did not accumulate BLOS1 at cell tips (Figure 3K) as what happened to LDLR (Figure 4B). It could be explained as the model: KIF13A delivered LDLR and then passed the cargo to BLOS1-KIF3. After that, did KIF3-BLOS1 continue to transport LDLR to plasma membrane or turn the cargo back to KIF13A? If it's the former one, it is surprising that overexpressing KIF3A and KIF3B didn't bring much LDLR to cell tips, though a lot BLOS1 was brought there (Figure 6—figure supplement 1). So how important of BLOS1-KIF3 is in KIF3A-mediated LDLR transport? Could the authors show KIF3A and LDLR (or TfR) distribution in BLOS1 KO cells like in Figure 4D? If the endogenous proteins were too low to detect in the KO cells, could the authors try cDNA constructs?

3) Compared to KIF13A, KIF3 has much more affinity to BLOS1 (Figure 5D), but not very strong associated with LDLR (Figure 5B). Could the author test how KIF3A-R will affect KIF13 and LDLR? Another experiment the authors could consider is quantifying cell pheriphery-localized LDLR with and without overexpression of KIF3A, KIF3B, and BLOS1, like in Figure 6—figure supplement 1.

4) Continuing to the first and second questions, the affinity between KIF13A and LDLR and between BLOS1 and KIF3A have been clearly shown, but the connection between these two chains was not directly addressed. In Figure 6, the authors addressed the defect of KIF3A and BLOS1 affected RE transport, but Rab11A was used as the RE marker. The authors did not show if the "active RE" that contains Rab11A is LDLR positive. In Figure 4H, KIF13A-R immobilized some of Rab11A, but there was still a lot of Rab11A not localized with KIF13A-R. In addition, dose Rab11A regulate the interactions of KIF13A-LDLR and/or BLOS1-KIF3A?

5) Is Figure 7C a 2D imagining? I think 3D imaging is a more proper way to show BLOS1 is at the intersection of microtubule tracks.

6) The authors found BLOS1 displayed different patterns in primary hepatocytes and in cell line *Hep* G2. In *Hep* G2 cells, BLOS1 localized at the microtubule intersections (Figure 7C), which supports the hypothesis, but in the primary hepatocytes, those tubule structures apparently are not at the intersections. Does that mean BLOS1 has other way(s) to associate with microtubules? Could the author characterize the BLOS1 puncta and tubule structures in the primary hepatocytes by co-staining BLOS1 with the cargo proteins and the motor proteins?

Reviewer #3:

1) The most concern of this reviewer is the discrepancy between data taken using primary hepatocytes and HepG2 cells. (i) LDLR vesicles were accumulated in the cell periphery in primary hepatocytes; however, they showed dispersed pattern in HepG2 cells (Figure 4A). (ii) Although KIF13A bound to BLOS1 in primary hepatocytes (and HEK293 cells) (Figure 3J,K), KIF13A did not bind to BLOS1 but bound to RE and LDLR to carry LDLR to the cell periphery in HepG2 cells (Figure 4D-J, Figure 5D). The authors concluded mainly from the data using HepG2 cells. Generally speaking, a primary culture is a more faithful tissue cell model than a cell line, and when their results do not match, those of primary culture is more reliable. In addition, the accumulation of LDLR to the cell periphery is not observed in HepG2 cells. Therefore, this reviewer believes that HepG2 cells are not suitable in analyzing the transport of LDLR from RE to the cell periphery. Considering the fact that primary hepatocytes actively transport LDLR to the cell periphery and that the authors have the advantage of using the primary hepatocytes lacking BLOS1, the authors should use primary hepatocytes instead of HepG2 cells.

2) Figure 4A: The authors showed the peripheral distribution of LDLR in the WT primary hepatocytes. Since BLOS1 KO primary hepatocytes showed reduced uptake of LDL via LDLR (Figure 2A,C), the authors should examine the distribution of LDLR in BLOS1 KO primary hepatocytes. The authors should also perform a series of transfection studies on WT and cKO primary hepatocytes (e.g. BLOS1 [rescue study on cKO], KIF13A [requirement of BLOS1 for KIF13A to bind to RE], KIF3A [requirement of BLOS1 for KIF3A to bind to LDLR] and so on).

3) Figure 4H/Figure 5C: In KIF13A-R (Figure 4H) or KIF3B-R (Figure 5C) over-expressed cells, KIF13A-R or KIF3B-R fully decorated the surface of non-physiologically bundled MTs. Therefore, the authors should exclude the possibility that microtubule-based motor protein cannot use such MTs as traffic rails. Authors should use another rigor mutant motor as a negative control, such as KIF5B-R or KIF16B-R, which does not interact with BLOS1.

4) Figure 4D, Figure 6E: Single over-expressed KIF13A in HepG2 cells showed vesicle pattern and accumulated to the cell periphery in Figure 4D, but showed tubular pattern in Figure 6E. Explain the reason why transfected KIF13A showed different pattern and distribution in HepG2 cells.

---

## [Author Response]

Essential revisions:1) The distribution of LDLR and KIF3A should be shown in BLOS1 KO cells.

See reply to point #2 of reviewer 3.

2) The co-localization of BLOS1 with LDLR, Rab 11A, and KIFs should be examined in primary hepatocytes.

See reply to point #6 of reviewer 2.

3) Figure 4A: The authors showed the peripheral distribution of LDLR in the WT primary hepatocytes. Since BLOS1 KO primary hepatocytes showed reduced uptake of LDL via LDLR (Figure 2A,C), the authors should examine the distribution of LDLR in BLOS1 KO primary hepatocytes. The authors should also perform a series of transfection studies on WT and cKO primary hepatocytes (e.g. BLOS1 [rescue study on cKO], KIF13A [requirement of BLOS1 for KIF13A to bind to RE], KIF3A [requirement of BLOS1 for KIF3A to bind to LDLR] and so on).

See reply to point #1 of reviewer 3.

4) Figure 4H/Figure 5C: In KIF13A-R (Figure 4H) or KIF3B-R (Figure 5C) over-expressed cells, KIF13A-R or KIF3B-R fully decorated the surface of non-physiologically bundled MTs. Therefore, the authors should exclude the possibility that microtubule-based motor protein cannot use such MTs as traffic rails. Authors should use another rigor mutant motor as a negative control, such as KIF5B-R or KIF16B-R, which does not interact with BLOS1.

See reply to point #3 of reviewer 3.

Reviewer #1:In this manuscript, authors reported BLOS1 is required for endosomal recycling of LDLR. Authors further shown BLOS1 act as adaptor protein for kinesin-2, moreover, they shown BLOS1 coordinate kinesin-3 and kinesin-2 for long-range transport of recycling endosome to pass microtubules intersections. This is an interesting study and overall, data are compelling. I have a few suggestions to further strength this study.1) The majority of conclusion are draw from imaging analysis on cells overexpressing various plasmids, although is not absolutely required, biochemical analysis using endogenous antibody can significantly strength their case, for example, author can purified the recycling endosome from control and BLOS1 KO mouse and compare the sorting of LDLR and the recruitment of kinesin-3 and kinesin-2 to recycling endosome.

We thank the reviewer for these suggestions. We agree that such biochemical analysis using endogenous antibody could strengthen our conclusion. However, only few endogenous antibodies worked in our hands. The purification of recycling endosome is a great idea to recapitulate our findings. Currently, we have not established the purification system.

2) Authors propose BLOS1 is an adaptor for kinesin-2, author need provide direct evidence by shown the recruitment of kinesin-2 to recycling endosome is reduced in BLOS1 KD or KO cells.

In our attempts to figure out the localization of kinesin-2 components in cells, we found that, in both *Hep* G2 cell lines (Figure 5E) and mouse primary hepatocytes, KIF3A was evenly distributed in cells (Figure 5—source data 1), without any enrichment in specific regions or puncta. Thus, the detection of recruitment of kinesin-2 to recycling endosomes is not possible even in control cells using immunofluorescent techniques or live-cell imaging.

3) The interaction between BLOS1 and kinesin-2 and kinesin-3 need to be confirmed by endogenous IP.

We agree with the reviewer. We did try to use recombinant BLOS1 protein and different synthetic BLOS1 peptides as antigens to prepare a polyclonal rabbit BLOS1 antibody but all failed. Neither commercial kinesin-2 nor kinesin-3 antibody for IP is available. Thus, endogenous IP for BLOS1 and kinesin-2 (or kinesin-3) could not be performed at the current stage.

Reviewer #2:In this manuscript, the authors described a defect of LDL clearance in liver-BLOS1-KO mice and found it was caused by reduced LDLR on cell surface due to the interrupted recycling endosome transport mediated by motor kinesin-3 and kinesin-2. BLOS1 served as the adaptor to switch the motors at microtubule intersections for recycling endosome long-range anterograde transport to cell periphery. The novelty of this manuscript is BLOS1-caused dyslipidemia and BLOS1 acts as an adaptor for kinesin-2, which could be new functions of BLOS1 independent of BLOC1 and BORC complexes. The manuscript was well written, and most figures were well organized to convey the information. To make the hypothesis more convincing, there are several concerns:1) The authors have shown PCSK9 level in the liver tissue of cKO mice (Figure 2—figure supplement 1B). PCSK9 is a secreted protein and initiates the function to LDLR in plasma. Could the authors also test PCSK9 level in the plasma?

Before detecting the PCSK9 level in liver tissue, we indeed tried to determine the plasma PCSK9 level using the same antibody that has been verified in literature (Abcam, ab31762). But no specific band could be detectable at the target molecular weight region using western blotting, which we thought may be due to the low concentration of PCSK9 in plasma compared to liver tissue. ELISA assay could be an alternative detection method, but we have not yet tried.

2) The authors have nicely shown LDLR is tightly associated with KIF13A (Figure 4C and E). Overexpressing KIF13A could accumulate the cargos LDLR and TfR at cell tips (Figure 3B, Figure 4B and D). It looks like KIF13A did not closely associate with BLOS1 (Figure 5D), and overexpressing KIF13A did not accumulate BLOS1 at cell tips (Figure 3K) as what happened to LDLR (Figure 4B). It could be explained as the model: KIF13A delivered LDLR and then passed the cargo to BLOS1-KIF3. After that, did KIF3-BLOS1 continue to transport LDLR to plasma membrane or turn the cargo back to KIF13A? If it's the former one, it is surprising that overexpressing KIF3A and KIF3B didn't bring much LDLR to cell tips, though a lot BLOS1 was brought there (Figure 6—figure supplement 1). So how important of BLOS1-KIF3 is in KIF3A-mediated LDLR transport? Could the authors show KIF3A and LDLR (or TfR) distribution in BLOS1 KO cells like in Figure 4D? If the endogenous proteins were too low to detect in the KO cells, could the authors try cDNA constructs?

Although without much evidence, we prefer the latter model in which KIF3-BLOS1 turn the cargo back to KIF13A after passing obstacles, since the former one can be excluded from our observations below. If KIF3-BLOS1 could function to transport LDLR to the plasma membrane, we should observe peripheral accumulation of LDLR after co-overexpression of KIF3A, KIF3B and BLOS1. But we found that, although some BLOS1 signals occurred at the cell periphery, the distribution of LDLR was not obviously compromised (Figure 6—figure supplement 1D). Thus, the former model is not supported by our observation. As shown in the reply to point #2 of reviewer 1, we found that KIF3A showed even distribution in both *Hep* G2 and mouse primary hepatocytes, it is difficult to observe and quantify the colocalization of KIF3A and LDLR (or TfR).

3) Compared to KIF13A, KIF3 has much more affinity to BLOS1 (Figure 5D), but not very strong associated with LDLR (Figure 5B). Could the author test how KIF3A-R will affect KIF13 and LDLR? Another experiment the authors could consider is quantifying cell pheriphery-localized LDLR with and without overexpression of KIF3A, KIF3B, and BLOS1, like in Figure 6—figure supplement 1.

We have showed that KIF3B-R did not affect the distribution of LDLR by immunofluorescence (Figure 5B). And, when observing the movement of KIF13A tubules after overexpression of KIF3B-R using live-cell imaging (see new Figure 6—video 4), similar repeated forward and backward movement which occurred after KIF3A knockdown could be observed. This phenomenon is consistent with our previous results. We showed in our manuscript that overexpression of KIF3A, KIF3B or BLOS1 didn’t significantly affect LDLR distribution like what KIF13A did. Even co-overexpression of KIF3A, KIF3B and BLOS1 did not obviously redistribute LDLR (see reply to the second comment). Thus, we thought KIF3A, KIF3B or BLOS1 is unlikely to influence the distribution of LDLR near cell periphery.

4) Continuing to the first and second questions, the affinity between KIF13A and LDLR and between BLOS1 and KIF3A have been clearly shown, but the connection between these two chains was not directly addressed. In Figure 6, the authors addressed the defect of KIF3A and BLOS1 affected RE transport, but Rab11A was used as the RE marker. The authors did not show if the "active RE" that contains Rab11A is LDLR positive. In Figure 4H, KIF13A-R immobilized some of Rab11A, but there was still a lot of Rab11A not localized with KIF13A-R. In addition, dose Rab11A regulate the interactions of KIF13A-LDLR and/or BLOS1-KIF3A?

We show here that LDLR colocalizes well with RAB11A in *Hep* G2 cells (Figure 4—figure supplement 1D), and most “active RE” is also LDLR positive (see new Figure 4—video 2). In our observation, most of Rab11A vesicles located along KIF13A-R tubules and were immobilized by KIF13A-R. We updated a newer version of Figure 4H and Figure 4-video 1 (see revised Figure 4 and new Figure 4—video 1), as the details of KIF3B-R rigor near the cell periphery were not well detected in the original movie. For the involvement of Rab11A in KIF13A-LDLR or BLOS1-KIF3A, we didn’t have any direct evidence. We suspect that Rab11A should play some role in the overall process of LDLR transport by KIF13A, since Delevoye et al. reported that Rab11A interact with KIF13A, and is required for the periphery distribution of overexpressed KIF13A (Delevoye et al., 2014).

5) Is Figure 7C a 2D imagining? I think 3D imaging is a more proper way to show BLOS1 is at the intersection of microtubule tracks.

Yes, Figure 7C is a super-resolution 2D image. According to the suggestion, we acquired z-stack super-resolution images to show the 3D localization of BLOS1 and microtubules (Figure 7—video 1). Since BLOS1 locates near microtubule intersections but not just at the microtubule intersection sites, and dynamically change 3D localization along microtubules (Figure 8—video 2), z-stack images may have the advantage of detecting the BLOS1 puncta near microtubule intersections in z dimension, which could not be obviously observed in 2D images.

6) The authors found BLOS1 displayed different patterns in primary hepatocytes and in cell line Hep G2. In Hep G2 cells, BLOS1 localized at the microtubule intersections (Figure 7C), which supports the hypothesis, but in the primary hepatocytes, those tubule structures apparently are not at the intersections. Does that mean BLOS1 has other way(s) to associate with microtubules? Could the author characterize the BLOS1 puncta and tubule structures in the primary hepatocytes by co-staining BLOS1 with the cargo proteins and the motor proteins?

We found that in primary hepatocytes, BLOS1 accumulates in both tubular and puncta structures, the puncta structure resembles what we observed in *Hep* G2 cells, and locates near microtubule intersections (Figure 7—figure supplement 1A,B). For the reason of BLOS1 tubules occurred in only primary hepatocytes, we thought that it may be related to the different pattern of LDLR distribution in these two kinds of cells. We found that the peripherally distributed LDLR in primary hepatocytes were mostly located on clathrin-coated vesicles (labeled by clathrin light chain A, CLTA), while only a portion of LDLR were colocalized with CLTA (Figure 4—figure supplement 1A, B), which means that LDLR in primary hepatocytes were recycled faster to the cell membrane to achieve their physiological role to recycle the majority of low-density lipoproteins in plasma. In our model, BLOS1 is used to overcome obstacles in the process of LDLR recycling. We suspect that the tubular distribution of BLOS1 in primary hepatocytes represents the dense distribution of BLOS1 along microtubules to ensure smooth and fast recycling of LDLR to the plasma membrane.

Reviewer #3:1) The most concern of this reviewer is the discrepancy between data taken using primary hepatocytes and HepG2 cells. (i) LDLR vesicles were accumulated in the cell periphery in primary hepatocytes; however, they showed dispersed pattern in HepG2 cells (Figure 4A). (ii) Although KIF13A bound to BLOS1 in primary hepatocytes (and HEK293 cells) (Figure 3J,K), KIF13A did not bind to BLOS1 but bound to RE and LDLR to carry LDLR to the cell periphery in HepG2 cells (Figure 4D-J, Figure 5D). The authors concluded mainly from the data using HepG2 cells. Generally speaking, a primary culture is a more faithful tissue cell model than a cell line, and when their results do not match, those of primary culture is more reliable. In addition, the accumulation of LDLR to the cell periphery is not observed in HepG2 cells. Therefore, this reviewer believes that HepG2 cells are not suitable in analyzing the transport of LDLR from RE to the cell periphery. Considering the fact that primary hepatocytes actively transport LDLR to the cell periphery and that the authors have the advantage of using the primary hepatocytes lacking BLOS1, the authors should use primary hepatocytes instead of HepG2 cells.

i) For the possible reason for the different distribution pattern of LDLR in *Hep* G2 cells and primary hepatocytes, we have explained in the reply to the point #6 of reviewer 2.

ii) We observed that BLOS1 tubules in primary hepatocytes locates on KIF13A-positive microtubules, and proposed that BLOS1 and KIF13A may function at same microtubule tracks. For the co-IP assay used to detect the interaction between BLOS1 and KIF13A was performed in HEK293T cells, so we think this interaction may be conserved among these cells. Considering that most of LDLR in primary hepatocytes locates on clathrin-coated vesicles, it is very difficult to catch every recycling step in such a fast recycling system. In addition, we observed that LDLR in *Hep* G2 cells partially colocalizes with clathrin-coated vesicles, early endosomes (Figure 4—figure supplement 1C) and recycling endosomes, suggesting that *Hep* G2 cells may be a suitable model to observe the LDLR recycling process.

2) Figure 4A: The authors showed the peripheral distribution of LDLR in the WT primary hepatocytes. Since BLOS1 KO primary hepatocytes showed reduced uptake of LDL via LDLR (Figure 2A,C), the authors should examine the distribution of LDLR in BLOS1 KO primary hepatocytes. The authors should also perform a series of transfection studies on WT and cKO primary hepatocytes (e.g. BLOS1 [rescue study on cKO], KIF13A [requirement of BLOS1 for KIF13A to bind to RE], KIF3A [requirement of BLOS1 for KIF3A to bind to LDLR] and so on).

We showed here that LDLR in BLOS1 KO primary hepatocytes showed similar periphery distribution (Figure 2—figure supplement 1C). After BLOS1 depletion, overexpressed KIF13A still could transport LDLR to the cell periphery, which indicates that BLOS1 is not required for KIF13A to bind to RE (Figure 4—source data 1). We thought that although BLOS1 is needed for KIF13A tubules to overcome obstacles, the large amount of overexpressed KIF13A could result in a continuous forward driving force, and once the obstacle has disappeared by the dynamic changing of microtubule architecture or other reasons, the cargoes transported by KIF13A will go on moving forward and finally reach cell periphery.

3) Figure 4H/Figure 5C: In KIF13A-R (Figure 4H) or KIF3B-R (Figure 5C) over-expressed cells, KIF13A-R or KIF3B-R fully decorated the surface of non-physiologically bundled MTs. Therefore, the authors should exclude the possibility that microtubule-based motor protein cannot use such MTs as traffic rails. Authors should use another rigor mutant motor as a negative control, such as KIF5B-R or KIF16B-R, which does not interact with BLOS1.

Here we showed that neither KIF3B-R nor KIF13A-R affected the movement of lysosomes (Figure 4—video 2 and Figure 5—video 2), which means that cargoes transport by other microtubule-based motor proteins could use these traffic rails.

4) Figure 4D, Figure 6E: Single over-expressed KIF13A in HepG2 cells showed vesicle pattern and accumulated to the cell periphery in Figure 4D, but showed tubular pattern in Figure 6E. Explain the reason why transfected KIF13A showed different pattern and distribution in HepG2 cells.

Actually, when KIF13A was overexpressed in cells, both tubular pattern and periphery accumulated pattern could be observed in one cell, which is the case in Figure 4D. Compared to fixed cells above, the tubular pattern of KIF13A will become more obvious in live-cell imaging, as shown in Figure 6E. This phenomenon has been reported by other literature (Delevoye et al., 2016; Delevoye et al., 2014). We think that the fixation and permeabilization processes in immunofluorescence microscopy may destabilize the tubular structure of KIF13A, thus results in less observed KIF13A tubular structures.